# A distinct cardiopharyngeal mesoderm genetic hierarchy establishes antero-posterior patterning of esophagus striated muscle

Glenda Comai[1,2†], Eglantine Heude[1,2,3†], Sebastian Mella[1,2], Sylvain Paisant[1,2], Francesca Pala[1,2,4], Mirialys Gallardo[5], Francina Langa[6], Gabrielle Kardon[5], Swetha Gopalakrishnan[1,2,7], Shahragim Tajbakhsh[1,2]*

[1]Department of Developmental and Stem Cell Biology, Institut Pasteur, Paris, France; [2]CNRS UMR 3738, Paris, France; [3]Department Adaptation du Vivant, CNRS/MNHN UMR 7221, Muséum national d'Histoire naturelle, Paris, France; [4]Laboratory of Clinical Immunology and Microbiology (LCIM), National Institutes of Health, Bethesda, United States; [5]Department of Human Genetics, University of Utah, Salt Lake City, United States; [6]Mouse Genetics Engineering Center, Institut Pasteur, Paris, France; [7]Institute of Biotechnology, HiLIFE, University of Helsinki, Helsinki, Finland

**Abstract** In most vertebrates, the upper digestive tract is composed of muscularized jaws linked to the esophagus that permits food ingestion and swallowing. Masticatory and esophagus striated muscles (ESM) share a common cardiopharyngeal mesoderm (CPM) origin, however ESM are unusual among striated muscles as they are established in the absence of a primary skeletal muscle scaffold. Using mouse chimeras, we show that the transcription factors *Tbx1* and *Isl1* are required cell-autonomously for myogenic specification of ESM progenitors. Further, genetic loss-of-function and pharmacological studies point to MET/HGF signaling for antero-posterior migration of esophagus muscle progenitors, where *Hgf* ligand is expressed in adjacent smooth muscle cells. These observations highlight the functional relevance of a smooth and striated muscle progenitor dialogue for ESM patterning. Our findings establish a *Tbx1-Isl1-Met* genetic hierarchy that uniquely regulates esophagus myogenesis and identify distinct genetic signatures that can be used as framework to interpret pathologies arising within CPM derivatives.
DOI: https://doi.org/10.7554/eLife.47460.001

*For correspondence:
shahragim.tajbakhsh@pasteur.fr

†These authors contributed equally to this work

Competing interests: The authors declare that no competing interests exist.

## Introduction

Evolution of vertebrates has been marked by the emergence of muscularized jaws that transitioned them from filter feeders to active predators (*Glenn Northcutt, 2005*). Considerable diversity in developmental origins and regulation of skeletal muscles point to important functional differences that remain unexplored. Muscles of the trunk originate from the segmented somites, whereas head muscles arise independently from the cardiopharyngeal mesoderm (CPM) located anterior to the somites (*Diogo et al., 2015*; *Sambasivan et al., 2011*). The specification of head and trunk muscles involves divergent genetic regulatory networks, to activate the bHLH myogenic regulatory factors (MRFs) Myf5, Mrf4, Myod and Myogenin that play crucial roles in governing striated muscle cell fate and differentiation (*Comai et al., 2014*; *Kassar-Duchossoy et al., 2005*; *Rudnicki et al., 1993*).

While somitic myogenic progenitors are regulated primarily by the *Pax3/Pax7* paired/homeodomain genes and *Myf5* that act genetically upstream of *Myod* (*Kassar-Duchossoy et al., 2005*;

*Relaix et al., 2005*; *Tajbakhsh et al., 1997*), cardiopharyngeal mesoderm progenitors, that colonize pharyngeal arches and form craniofacial and some neck muscles, are regulated by a *Pax3*-independent regulatory network (*Heude et al., 2018*; *Sambasivan et al., 2011*). CPM progenitors specified by *Tbx1* and *Isl1* genes are bipotent as they form branchiomeric subsets of head/neck muscles as well as the second heart field (*Diogo et al., 2015*; *Kelly et al., 2004*; *Lescroart et al., 2015*; *Sambasivan et al., 2009*). *Tbx1* acts together with *Myf5* to assure myogenic fate (*Harel et al., 2009*; *Kelly et al., 2004*; *Nathan et al., 2008*; *Sambasivan et al., 2009*). In *Tbx1*-null embryos, the first pharyngeal arch is hypoplastic and posterior pharyngeal arches do not form, resulting in variably penetrant defects of masticatory muscles and absence of muscles derived from more posterior arches including those of the larynx and esophagus (*Gopalakrishnan et al., 2015*; *Heude et al., 2018*; *Kelly et al., 2004*; *Lescroart et al., 2015*). *Tbx1* exerts cell-autonomous and non-autonomous roles as conditional deletion of *Tbx1* in CPM and pharyngeal endoderm phenocopies the pharyngeal arch and cardiac outflow tract phenotype of the null mutant (*Arnold et al., 2006*; *Kelly et al., 2004*; *Zhang, 2006*). On the other hand, the functional role of *Isl1* in CPM specification remains unknown due to early embryonic lethality of *Isl1*-null mutants (by E10.5) that exhibit cardiac deficiencies (*Cai et al., 2003*; *Harel et al., 2009*; *Nathan et al., 2008*). Thus, due to the severe phenotypes observed in the mouse, the epistatic relationship between *Tbx1* and *Isl1* and their cell-autonomous roles during CPM-derived muscle specification remain unclear.

Recent studies by us and others showed that CPM progenitors generate diverse myogenic subpopulations at the transition zone between head and trunk (*Diogo et al., 2015*; *Gopalakrishnan et al., 2015*; *Heude et al., 2018*; *Lescroart et al., 2015*; *Schubert et al., 2019*; *Tabler et al., 2017*). Whether CPM muscle derivatives form a homogeneous group specified by a unique gene regulatory network is unknown. We have previously shown that esophagus striated muscles (ESM) arise from the CPM and exhibit several features that are distinct from other striated muscles in the organism. Notably, ESM formation initiates in the fetus, thus embryonic myogenesis which generates primary myofibers that act as scaffolds for secondary (fetal) myofibers does not take place (*Gopalakrishnan et al., 2015*). As the esophagus is the only site identified to date that undergoes this unusual patterning, this raises the issue of what cell type (s) pattern the ESM.

The mammalian esophagus is composed of both striated and smooth muscle layers, which have a distinct developmental origin (*Gopalakrishnan et al., 2015*; *Krauss et al., 2016*; *Rishniw et al., 2003*; *Zhao and Dhoot, 2000a*). Postnatal maturation of the esophagus striated musculature involves proximo-distal replacement of smooth muscle by as yet elusive mechanisms (*Krauss et al., 2016*). Although smooth muscle and other mesenchymal cells are in close proximity to ESM progenitors as they undergo lineage commitment and differentiation, how the latter are patterned in the absence of primary myofibers remains unknown. It has been proposed that the esophagus smooth muscle may provide a scaffold for laying down ESM myofibers, however it is unclear to what extent this differs from other sites in the organism where striated muscles play this role (*Gopalakrishnan et al., 2015*; *Zhao and Dhoot, 2000a*).

Perturbations of esophagus function lead to dysphagia and other pathophysiological disorders that impair swallowing and transfer of bolus to the stomach (*Sheehan, 2008*). ESM share a common origin with branchiomeric head muscles in which *Tbx1* and *Isl1* act as upstream regulators of myogenic specification (*Gopalakrishnan et al., 2015*; *Heude et al., 2018*). In *Tbx1*-null embryos, *Isl1*-derived myogenic cells fail to seed the anterior esophagus, suggesting that *Tbx1* acts genetically upstream of *Isl1* in ESM progenitors (*Gopalakrishnan et al., 2015*). Initially, CPM-derived progenitors are seeded at the bottom of the oropharyngeal cavity by E13.5. Then, *Isl1*-derived ESM progenitors colonize the esophagus by posterior migration and differentiation until the third week of postnatal growth (*Gopalakrishnan et al., 2015*; *Romer et al., 2013*). How these *Isl1*-derived progenitors colonize the structure while restricting premature differentiation remains unknown.

Muscle progenitors undergo short-range displacement or long-range migration for establishing skeletal muscles, as exemplified by myotomes and limbs, respectively. Progenitors originating from ventral somites delaminate and emigrate to distal sites to give rise to trunk, limb and tongue muscles (*Bladt et al., 1995*; *Brand-Saberi et al., 1996*; *Dietrich et al., 1999*). This process is regulated by the tyrosine kinase receptor MET, expressed in migratory progenitors, and its ligand Scatter Factor/Hepatocyte Growth Factor (SF/HGF) expressed in mesenchymal cells along the migratory route (*Bladt et al., 1995*; *Brand-Saberi et al., 1996*; *Dietrich et al., 1999*). Knockout of either *Met* or *Hgf* in mice results in the absence of hypaxial muscles including limb muscles, diaphragm and the

tip of the tongue (*Bladt et al., 1995*; *Dietrich et al., 1999*; *Maina et al., 1996*; *Prunotto et al., 2004*). Although second (hyoid) arch-derived muscles are affected in *Met* KO mice (*Prunotto et al., 2004*), a role for MET/HGF signaling in establishing other CPM muscles including those in the larynx and esophagus has not been reported.

In the present study, we used mouse chimeras to circumvent lethality issues and assess the cell-autonomous roles of *Tbx1* and *Isl1* in ESM progenitors. Using genetic loss-of-function and pharmacological inhibition approaches, we show that MET/HGF signaling is critical for ESM patterning, but not for the establishment of adjacent laryngeal muscles. These studies unveil an unexpected *Tbx1/Isl1/Met* genetic hierarchy operating within a CPM-muscle group, thereby identifying distinct genetic signatures for these evolutionarily conserved mesodermal derivatives.

## Results

### Requirement of *Tbx1* and *Isl1* in ESM specification

We showed previously that *Tbx1*-null embryos lack ESM, wherein *Isl1*-derived myogenic progenitors fail to colonize and pattern the esophagus (*Gopalakrishnan et al., 2015*). The absence of seeding of *Isl1*-derived ESM progenitors in the anterior esophagus of *Tbx1*-null mice could originate from cell-autonomous or non-autonomous defects. To distinguish between these possibilities, we generated two types of chimeric embryos to explore the epistatic relationship between *Tbx1* and *Isl1* during ESM formation. Embryonic chimeras are well-established tools that have provided key insights into the tissue-specific requirement of genes during mammalian development (*Tam and Rossant, 2003*).

We first generated chimeras by injection of *Isl1^{lacZ}* (KI) ES cells (*Sun et al., 2007*) in *Tbx1^{-/-}* and control (*Tbx1^{+/-}*) blastocysts to determine if *Tbx1/Isl1*-positive cells can colonize the esophagus in a *Tbx1*-null environment. Here, β-galactosidase (β-gal) expression is under the control of the *Isl1* promoter to trace the ES-derived cells in vivo (*Figure 1—figure supplement 1A,B*). All *Tbx1*-null chimeric embryos analyzed between E14.5 and E15.5 lacked thymus glands (n=9) and 77% of them were edemic and lacked the outer ear pinna indicating that the contribution of *Isl1^{lacZ}* ES cells (*Tbx1^{wild-type}*) was not extensive enough to fully rescue the *Tbx1* knockout phenotype (*Figure 1—figure supplement 1C,D*). Whole mount X-gal staining showed that 5/5 chimeric *Tbx1^{-/-}* embryos analyzed contained β-gal+ cells in the esophagus (*Figure 1—figure supplement 1E*), though to variable extent in individual embryos when compared to heterozygous controls. Apart from its expression in ESM progenitors, *Isl1* is expressed in peripheral neurons (*Pfaff et al., 1996*) and in the pharyngeal and esophageal epithelium (*Cai et al., 2003*; *Harel et al., 2009*; *Nathan et al., 2008*). Therefore, we analyzed tissue sections to assess β-gal expression pattern at the cellular level. We observed that β-gal+ cells were present within the smooth muscle layers of the esophagus of *Tbx1*-null embryos (8/8 chimeras analyzed), and colocalised within Tnnt3+/Tuj1- (myogenic/non-neurogenic) cells (*Figure 1—figure supplement 1F,G*). While ESM colonization in chimeric *Tbx1^{-/-}* embryos appeared to be less efficient than in controls (determined by number of β-gal+ cells/section and Tnnt3+ muscle area/section), the relative number of β-gal+ cells/Tnnt3+ muscle area was non-significantly altered (*Figure 1—figure supplement 1H–J*). Taken together, these data indicate that *Isl1^{lacZ}* ES cells can colonize an overall *Tbx1*-null esophageal environment suggesting cell autonomous potential of Tbx1+/Isl1+ progenitors to seed and pattern the ESM. Of note, cells expressing lower levels of β-gal were present in the esophagus epithelia and connective tissue layers of both control and chimeric *Tbx1*-null embryos suggesting that, in these cells, expression from the endogenous *Isl1* locus was downregulated, and therefore the extent of the contribution of *Isl1^{lacZ}* ES cells (*Tbx1^{wildtype}*) cannot be unambiguously assessed. To circumvent this issue, we generated a second series of chimeras to address the intrinsic role of *Isl1* during fetal esophagus myogenesis.

To bypass the early embryonic lethality of *Isl1*-null embryos (*Cai et al., 2003*), we generated chimeric fetuses by injection of *Isl1*-null ES cells into wildtype (WT) mouse blastocysts. We targeted *Isl1*-null and control (WT) ES cells with a constitutive *lacZ* expression cassette (*pCAG-nlacZ*; nuclear β-gal activity) to trace ES cell derivatives ubiquitously and independently of *Isl1* expression (*Figure 1A–B*). Macroscopic examination of chimeras at E16.5 did not reveal obvious developmental defects in *Isl1*-null chimeras compared to controls. Immunostainings on sections were then performed to evaluate the contribution of *Isl1*-null (ES: *Isl1^{-/-}*; *nlacZ*) and control (ES: WT; *nlacZ*) β-gal+ ES-derived cells to the esophagus myogenic population. For reference, contribution of β-gal+ ES-

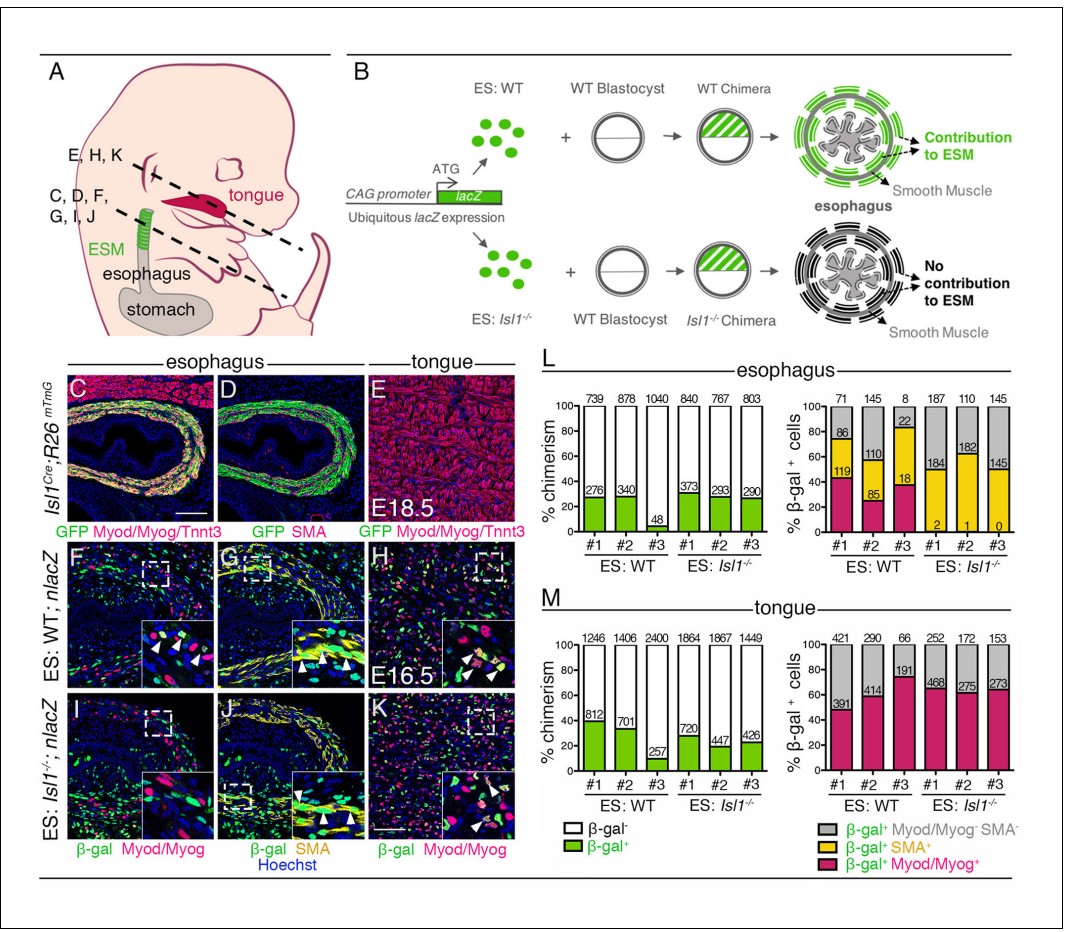

**Figure 1.** Cell-autonomous role of *Isl1* in esophagus myogenic progenitors. (**A**) Structures and levels analyzed in the study. (**B**) Schematic summary of the chimera experiment. (**C–E**) Immunostainings on transverse cryosections of a E18.5 *Isl1^Cre^;R26^mTmG^* fetus for the GFP reporter, Myod/Myog/Tnnt3 (myogenic markers) and SMA (smooth muscle actin) in the esophagus and tongue. Note that *Isl1*-derivatives include the esophagus striated muscle but not the esophagus smooth muscle layers and tongue muscle (n = 2). (**F–K**) Immunostainings on transverse cryosections of E16.5 WT (ES: WT;*nlacZ*) and *Isl1^-/-^* (ES: *Isl1^-/-^*;*nlacZ*) chimeras for the β-gal reporter, Myod/Myog (myogenic markers) and SMA (smooth muscle actin) in the esophagus and tongue (n = 3 each condition). Insets (bottom, right), higher magnifications. White arrowheads indicate examples of β-gal colocalization with SMA or Myod/Myog. (**L–M**) Percentage of chimerism and of β-gal+ cell contribution to the indicated populations in the esophagus and tongue of WT (ES: WT) and *Isl1^-/-^* (ES: *Isl1^-/-^*) chimeras (n = 3 each condition, #1–3; three different section levels scored). The total number of cells counted on three different section levels are reported in columns. Note that the *Isl1^-/-^* ES-derived cells do not form ESM progenitors but contribute to both esophagus smooth muscle layers and tongue. Scale bars: C, 100 μm; K, 50 μm.

DOI: https://doi.org/10.7554/eLife.47460.002

The following figure supplement is available for figure 1:

**Figure supplement 1.** *Isl1^nlacZ/+^* cells colonize the esophagus of chimeric *Tbx1* mutants.

DOI: https://doi.org/10.7554/eLife.47460.003

derived cells was compared with Isl1 lineage tracing (*Isl1^Cre/+^;R26^mT/mG/+^* embryos), whereby GFP+ *Isl1*-derived CPM cells contribute to the esophagus myogenic population (Myod/Myog/Tnnt3+), but not to the esophagus smooth muscle (SMA+) and striated muscle of tongue that develop in an *Isl1*-independent context (*Figure 1C–E*). We then quantified the percentage of chimerism and percentage of β-gal+ cells in the esophagus SMA+ and Myod/Myog+ populations and in myogenic cells of the tongue (n = 3, *Figure 1F–K*). In both *Isl1*-null and control chimeras, the overall percentage of chimerism in the tongue was similar to that observed in the muscularised layers of the esophagus (*Figure 1L–M*, left panels). In the esophagus of control chimeras, β-gal+ cells gave rise to both

SMA+ (31–46%) and Myod/Myog+ populations (25–43%) (*Figure 1F,G,L*). In contrast, *Isl1*-null/β-gal + cells were excluded from the esophagus Myod/Myog+ cells (*Figure 1I,J,L*), whereas they contributed to a similar extent to esophagus smooth muscle and tongue myogenic cells in both *Isl1*-null and control chimeras. These results show that *Isl1* is necessary cell-autonomously for progenitor cells to adopt a myogenic cell fate in the esophagus (*Figure 1B*).

## Spatiotemporal activation of the ESM myogenic program

ESM development occurs in a biphasic mode, with initial seeding of *Isl1*-positive myogenic progenitors at the anterior esophagus followed by anterior-posterior migration and differentiation that proceeds to postnatal stages (*Gopalakrishnan et al., 2015*). Isl1 is known to maintain cells in an undifferentiated state in branchiomeric muscle progenitors (*Cai et al., 2003*; *Nathan et al., 2008*). To determine the expression level of *Isl1* in ESM progenitors relative to lineage committed MRF

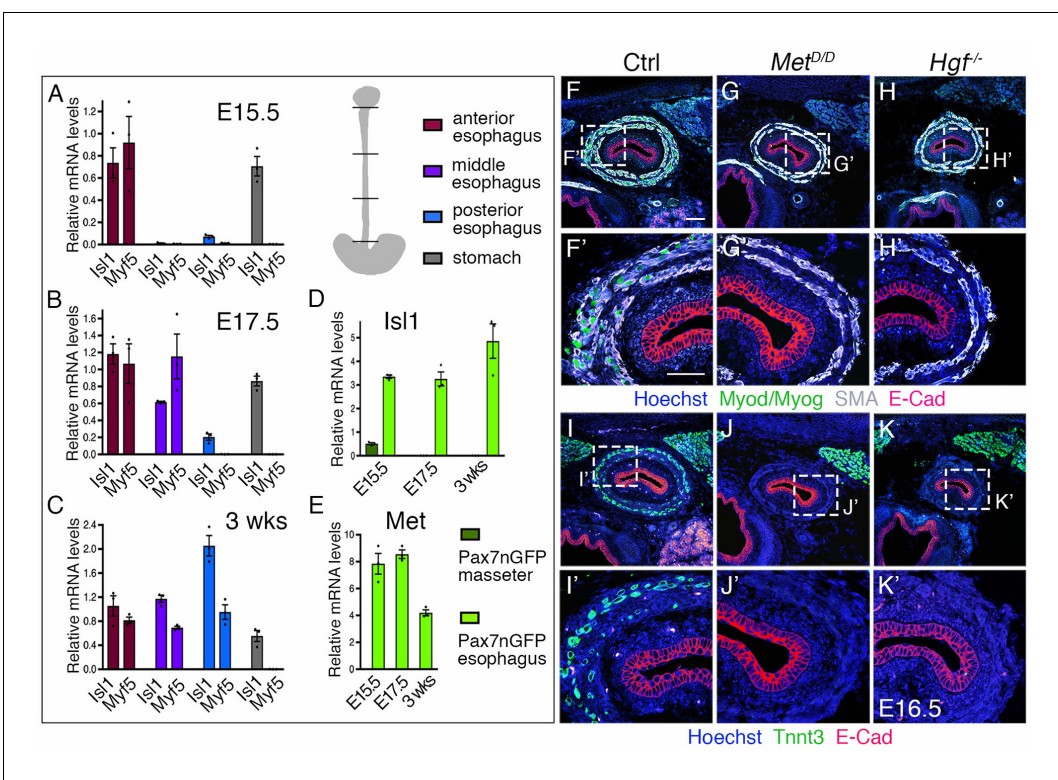

**Figure 2.** Regulation of esophagus striated muscle patterning involves MET/HGF signaling. (**A–C**) RT-qPCR analysis for *Isl1* and *Myf5* at E15.5 (**A**), E17.5 (**B**) and 3 weeks postnatal (**C**) in different esophagus portions and stomach as indicated in the schematic view (top, right). The low level of *Isl1* expression in the posterior esophagus at fetal stages might reflect contamination from the stomach at the esophagus interface (n = 3 each condition). (**D–E**) RT-qPCR analysis for *Isl1* and *Met* at E15.5, E17.5 and 3 weeks postnatal in *Tg:Pax7-nGFP*+ cells isolated by FACS from the masseter or esophagus. All data points are plotted and presented as the mean ± SEM (error bars) (n = 3 each condition). (**F–H**) Immunostainings on transverse cryosections of E16.5 control, *Met^{D/D}* and *Hgf^{-/-}* fetuses for Myod/Myog (myogenic progenitors) and SMA (smooth muscle actin). E-Cad labels the esophagus lumen epithelium. Higher magnifications are shown in (**F'–H'**) (n = 3 each condition). (**I–K**) Immunostainings on transverse cryosections of E16.5 control, *Met^{D/D}* and *Hgf^{-/-}* fetuses for Tnnt3 (myofiber marker) and E-Cad. Higher magnifications are shown in (**I'–K'**). Note the absence of ESM formation in both *Met* and *Hgf* mutants (n = 3 each condition). Scale bars: F, 100 µm; F', 50 µm.

DOI: https://doi.org/10.7554/eLife.47460.004

The following figure supplements are available for figure 2:

**Figure supplement 1.** *Met* and *Hgf* expression along the developing esophagus.
DOI: https://doi.org/10.7554/eLife.47460.005

**Figure supplement 2.** Phenotype of *Met* and *Hgf* mutants.
DOI: https://doi.org/10.7554/eLife.47460.006

genes, we performed RT-qPCR analysis at key stages of ESM development: at E15.5 and E17.5 when one third and two thirds of the esophagus is populated by ESM progenitors, respectively; then at 3 weeks postnatally when the entire esophagus is muscularised. *Isl1* and *Myf5* transcripts were detected in the anterior, middle and posterior esophagus as ESM progenitors colonize the structure from E15.5 to 3 weeks postnatally (*Figure 2A–C*). *Isl1* expression was also detected in the stomach, as already described for the gastric epithelium (*Das and May, 2011*). We next performed RT-qPCR analysis for *Isl1* in myogenic cells isolated from *Tg:Pax7-nGFP* mice where Pax7+ progenitors can be isolated from mid-embryonic stages (*Sambasivan et al., 2009*) (*Figure 2D*). *Isl1* was expressed in Pax7-nGFP+ esophagus progenitors, whereas expression was low or undetectable in those isolated from the masseter at all stages analyzed. Therefore, *Isl1* expression is maintained in myogenic progenitors throughout ESM development.

We then asked what molecular pathways would guide ESM progenitors to undergo A-P migration. Given the key role for MET/HGF signaling during delamination and long distance migration of hypaxial muscle progenitors (*Bladt et al., 1995*; *Dietrich et al., 1999*; *Maina et al., 1996*; *Prunotto et al., 2004*), we performed RT-qPCR analysis for *Met* in isolated Pax7-nGFP+ myogenic cells of the esophagus, as well as for *Met*, *Hgf* and *SMA* (as a landmark for the smooth muscle scaffold) in anterior, middle and posterior whole esophagus portions. Notably, Pax7-nGFP+ ESM progenitors showed transcript abundance of *Met* at fetal stages and lower expression levels postnatally when ESM colonization was complete (*Figure 2E*). Along the anterior-posterior axis, *Met* levels were higher in the anterior esophagus portion at E13.5 and E15.5, and become upregulated also in the middle part by E17.5 (*Figure 2—figure supplement 1A*). *Hgf* levels seemed initially constant along the esophagus length at E13.5, but appeared downregulated anteriorly in parallel with SMA at later stages (*Figure 2—figure supplement 1B,C*). Taken together, these data suggest that MET/HGF signaling might be implicated in A-P migration of ESM progenitors from fetal to postnatal stages.

## Severe loss of ESM in *Met and Hgf* mutants

To address the role of MET/HGF signaling during ESM formation, we examined *Met*^D/D^ and *Hgf*-null mutants (*Maina et al., 1996*; *Schmidt et al., 1995*). We first analyzed the esophagus phenotype of *Met* and *Hgf* mutants at E16.5 by immunostainings on tissue sections for early myogenic and myofiber markers (Myod/Myog/Tnnt3), for smooth muscle (SMA) and lumen epithelium (E-Cad) markers. Interestingly, *Met*^D/D^ and *Hgf*-null fetuses showed absence of striated muscles in the esophagus, while the smooth muscle layers and lumen epithelium appeared unaffected (*Figure 2F–K*). As expected, these mutants lacked limb muscles typical of the *Met*^D/D^ and *Hgf*-null phenotypes (*Figure 2—figure supplement 2A–F*). However, RT-qPCR analysis in the esophagus and limb of *Met* mutants at E15.5 revealed a decrease but not loss of *Isl1* and *Myf5* expression compared to absence of *Pax7* and *Myf5* observed at limb level (*Figure 2—figure supplement 2G*).

Given this observation, we investigated whether myogenic cells are present at the anterior-most part of the esophagus and in adjacent *Isl1*-derived muscles in the *Met*^D/D^ fetuses (*Figure 3A–C*). Analysis on tissue sections revealed that the number of Isl1+ cells in the upper esophagus of E13.5 controls and *Met* mutants was not significantly different (n = 3) (*Figure 3A–B'*, *Figure 3—figure supplement 1A–A''*). Thus, the myogenic progenitors had seeded the anterior esophagus in mutant embryos similarly to controls at this initial stage. Moreover, analysis of *Met*^D/+^ ; *Myf5*^nlacZ/+^ (control) and *Met*^D/D^ ; *Myf5*^nlacZ/+^ (mutant) esophagi at E15.5 (*Figure 3—figure supplement 1B,C*) and E17.5 (*Figure 3D,E*; *Figure 3—figure supplement 1D,E*) showed that Myf5+ myogenic cells were also present in the anterior-most portion of the esophagus in the mutant, whereas colonization had proceeded posteriorly only in the controls. Of note, the neuronal and smooth muscle lineages were present and patterned in mutants (*Figure 3D,E*).

Next, we investigated the fate of the sporadic myogenic cells remaining in the anterior esophagus of *Met*-null fetuses. To this end, we combined the *Met*^D/D^ mutant with *Isl1* lineage tracing. Analysis of E14.5 *Met*^D/+^; *Isl1*^Cre/+^;R26^mT/mG/+^ control embryos showed that mGFP+ mononucleated cells were abundant in between myofibers (*Figure 3F*). However, very few mononucleated mGFP+ cells were detected between the residual myofibers in the anterior-most part of the esophagus in the *Met* mutant (*Figure 3G*). Strikingly, the adjacent *Isl1*-derived laryngeal and pharyngeal muscles were unaffected in the *Met*-null fetuses (*Figure 3C,C',D–G*; *Figure 3—figure supplement 1B–E*). Therefore, these observations indicate that MET/HGF loss of function affect only a subset of posterior CPM-derived progenitors that are critical for colonization of the esophagus but not for the

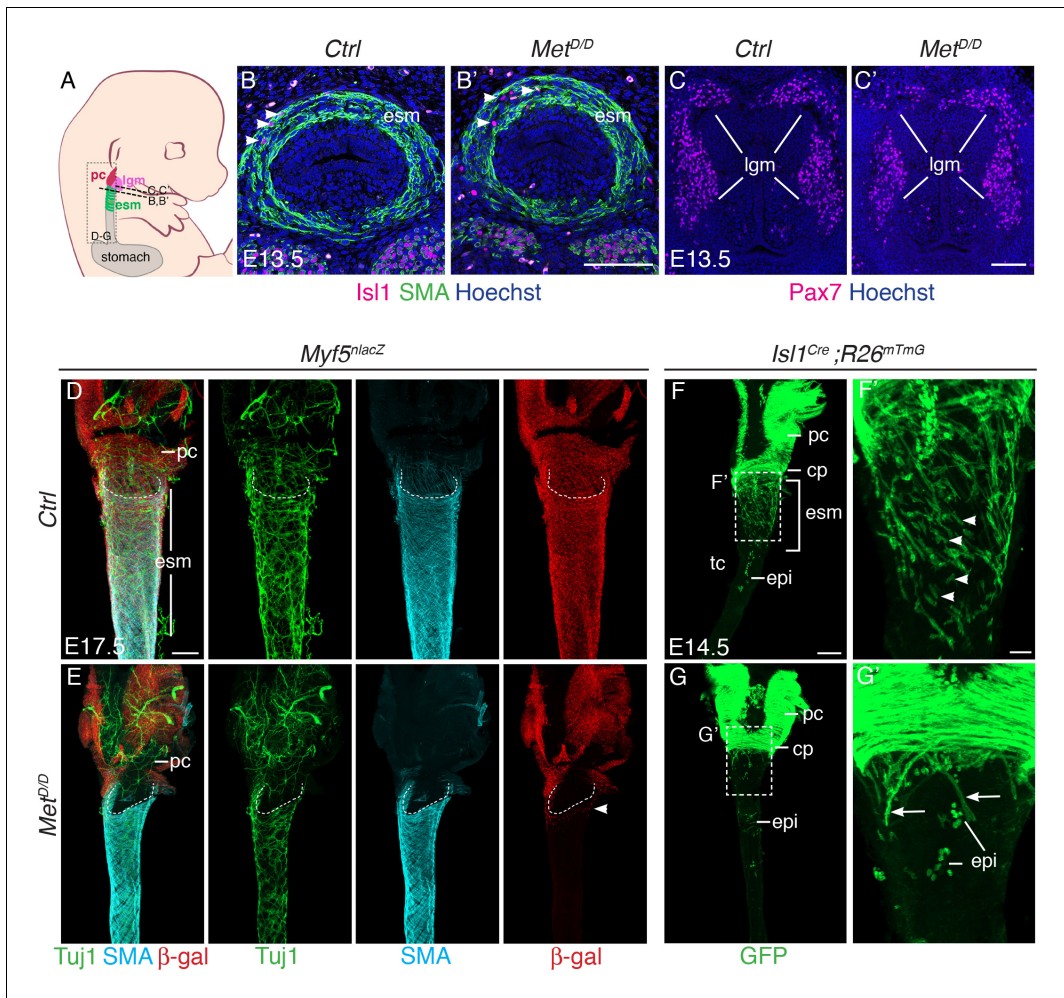

**Figure 3.** Isl1 progenitors are present anteriorly in the esophagus of $Met^{D/D}$ mutants. (**A**) Structures and levels analyzed in the study. (**B,B'**) Immunostainings on transverse cryosections of E13.5 control and $Met^{D/D}$ embryos for *Isl1* expressing progenitors (white arrowheads) and smooth muscle cells (SMA) in the esophagus (n = 2). (**C,C'**) Immunostainings on transverse cryosections at the laryngeal level of E13.5 control and $Met^{D/D}$ embryos for *Pax7* (n = 2). (**D–G**) Whole mount immunostaining of the upper esophagus of Met mutant and control embryos. (**D,E**) Ventral views of E17.5 esophagi stained for Tuj1 (neurons), SMA (smooth muscle actin) and β-gal ($Myf5^{nlacZ}$ reporter positive progenitors). White-dotted lines outline the shape of the esophagus entry. White arrowheads point to $Myf5^{nlacZ/+}$ progenitor cells present in the upper esophagus in the mutant. (**F,G**) Dorsal views of E14.5 stained for GFP (*Isl1* lineage tracing). *Isl1*-derived muscle progenitors remain largely as mononucleated cells in the control (F,F', white arrowheads) while GFP+ fibers are mostly seen in the anterior esophagus of the mutant (G,G', arrows) (n = 2 each condition). cp, cricopharyngeous muscle; epi, epithelial *Isl1*-derived cells; esm, esophagus striated muscle; lgm, laryngeal muscles; pc, pharyngeal constrictor. Scale bars: B', C', F', 50 μm; D, F, 200 μm.

DOI: https://doi.org/10.7554/eLife.47460.007

The following figure supplement is available for figure 3:

**Figure supplement 1.** Myogenic cells are present anteriorly in the esophagus of $Met^{D/D}$ mutants.

DOI: https://doi.org/10.7554/eLife.47460.008

development of adjacent *Isl1*-lineage-derived muscles. Taken together, these results indicate that *Met* acts downstream of *Isl1* in the molecular hierarchy of ESM formation and that MET/HGF signaling is not implicated in initial seeding of *Isl1*-derived progenitors in the anterior esophagus, but rather during the second phase of myogenic migration. This is in agreement with the biphasic mode of ESM development postulated previously (*Gopalakrishnan et al., 2015*).

## Requirement of MET/HGF signaling for A-P migration of ESM progenitors

To rule out the possibility that the defect in ESM formation in *Met* mutants is due to increased cell death in the anterior-most portion of the esophagus as opposed to aberrant migration of ESM cells, we tested if Isl1+ progenitors undergo apoptosis at the seeding stage in *Met* mutant embryos. TUNEL analysis in E13.5 controls and *Met* mutants revealed that Isl1+ progenitors in the upper esophagus were not apoptotic at this stage (*Figure 4—figure supplement 1A,A'*).

Next, we investigated the role of MET in the migration of mGFP+ cells in an ex vivo esophagus explant culture system once they had colonized the upper esophagus (*Figure 4A*). To this end, we employed static cultures and time-lapse confocal microscopy on E14.5 *Isl1^{Cre};R26^{mT/mG}* esophagus in combination with two selective ATP-competitive inhibitors of MET, PF-0417903 and MGCD-265 or DMSO as control (*Figure 4B–D*). On control static cultures (followed up to 24 hr) and time-lapse imaging (up to 14 hr), we observed a mGFP+ mononucleated cell front that remained throughout the entire culture period (*Figure 4—figure supplement 1B*, *Figure 4B*, *Figure 4—video 1*). In addition, time-lapse movies showed that mGFP+ cells explored the esophagus scaffold repeatedly changing their direction of migration, but had a net movement posteriorly towards the stomach (*Figure 4E*, *Figure 4—video 1*). Upstream of the mononucleated cell front, mGFP+ cells also migrated posteriorly in between forming fibers (*Figure 4—video 1*). In contrast, upon addition of MET inhibitors, mGFP+ cells progressed less towards the posterior end (*Figure 4C,D*, *Figure 4—videos 2 and 3*) and had shorter cell trajectories (*Figure 4F–H*). Quantification of migration parameters revealed that in presence of MET inhibitors, mGFP+ cells had a reduced velocity, displacement, efficiency, and net velocity compared to control cultures (*Figure 4I–L*). Interestingly, mGFP+ fibers appeared rapidly in the inhibitor treated cultures in positions where the cell density appeared higher (*Figure 4C,D*, *Figure 4—figure supplement 1C*), a phenotype that resembled *Met*-null embryos (*Figure 3G*, *Figure 3—figure supplement 1E*). To confirm this, we examined the proliferation and differentiation status of ESM myogenic cells in vivo by EdU labelling. Analysis of E14.5 and E15.5 embryos showed that myogenic cells in the esophagus of *Met^{D/D}* mutant embryos have a proliferation rate that was one third of controls (45,6% for Ctrl; 12,5% for Mutant at E14.5, *Figure 4—figure supplement 1D*) and a higher predisposition to differentiation as assessed by Myogenin expression (33% for Ctrl; 66% for Mutant at E14.5, *Figure 4—figure supplement 1E*).

In summary, our ex vivo and in vivo analyzes indicate that MET/HGF signaling promotes A-P migration of ESM myogenic progenitors during fetal development and possibly maintains *Isl1*-derived progenitors in an undifferentiated state once they have colonized the upper esophagus.

## Cellular relationships of *Isl1* and *Met/Hgf* expression in ESM progenitors

We then decided to examine in detail the expression pattern of *Hgf* and *Met* in relation to *Isl1*-derived progenitors. Owing to the limited diffusion efficiency of HGF in vivo, both receptor and ligand expressing cells are expected to be found in close proximity to each other (*Dietrich et al., 1999*). Whole mount in situ hybridization (RNAscope) on *Isl1^{Cre};R26^{mT/mG}* embryos showed that *Hgf* transcripts were present along the entire length of the esophagus, preceeding and following the myogenic cell front, at the seeding stage (E13.5) and during myogenic migration (E14.5) (*Figure 5—figure supplement 1A,B*). In situ hybridization on sections revealed that *Hgf* was expressed adjacent to mGFP+ cells in a bilayered concentric pattern overlapping with the smooth muscle layers of the esophagus (*Figure 5A,B*; *Figure 5—figure supplement 1*). Initially at E14.5, although a bilayered pattern of *Hgf* is observed, myogenic cells colonized exclusively the outer smooth muscle layer (*Figure 5A*). At E16.5, expression of *Hgf* appeared to be downregulated anteriorly in the outer layer (level 1; *Figure 5—figure supplement 1E*), concomitant with the described regression of smooth muscle in the anterior esophagus from late fetal stages (*Figure 2—figure supplement 1B–C*) (*Zhao and Dhoot, 2000a*). At this stage, myogenic cells started to colonize the inner layer anteriorly (*Zhao and Dhoot, 2000b*), and this corresponded to downregulation of *Hgf* expression in the outer layer (*Figure 5—figure supplement 1E–E''*). Quantifications of the amount of *Hgf* transcripts per smooth muscle area corroborated these observations and revealed uniform levels between the myogenic cell front and more posterior levels (E13.5 and E14.5, levels 1–3; E16.5, levels 2–4; *Figure 5—figure supplement 1F*). Altogether, our data indicate that the smooth muscle layers of the

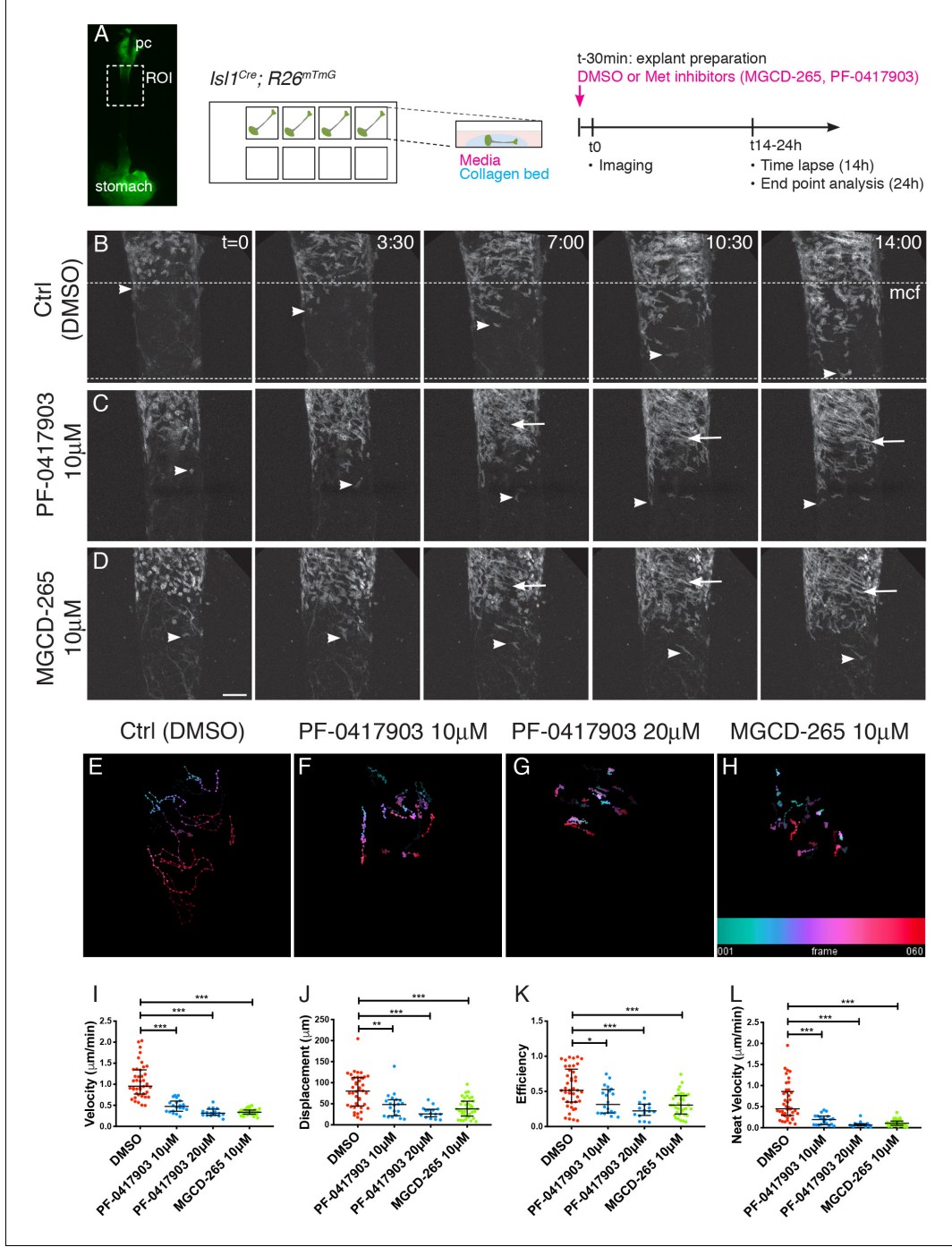

**Figure 4.** MET/HGF signaling is required for migration of *Isl1*-derived myogenic progenitors. (**A**) Macroscopic view of *Isl1^{Cre};R26^{mT/mG}* E14.5 dissected esophagus used for explant culture and live imaging. Esophagi were placed in collagen beds in individual Ibidi wells. MET inhibitors (MGCD-265, PF-0417903) or control (ctrl, DMSO) were added to explants 30 min before imaging. Explants were kept for 14 hr for live imaging (with an image taken every 12–15 min) or 24 hr for analysis at fixed time points (***Figure 4—figure supplement 1***). (**B–D**) Maximum projection of time series from a time-lapse experiment of esophagi explant culture in the presence of DMSO (**B**), 10 μM PF-0417903 (**C**) or 10 μM MGCD-265 (**D**). White arrowheads point to Isl1-derived progenitor cells present at the mononucleated cell front (mcf). White arrows highlight the high numbers of fibers that appear progressively in the inhibitor condition. Time (**t**) is indicated in hours. Dotted lines show the overall advancement of the mcf in the control condition. (**E–H**) Temporal color coded 2D images of GFP+ cell trajectories tracked in the time lapse movies in control and inhibitor treated explant cultures (related to ***Figure 4—videos 1***, ***2*** and ***3***). (**I–L**)

*Figure 4 continued on next page*

*Figure 4 continued*

Quantification of cell velocity (in μm/min; I), displacement (μm, the length of the resultant vector between ti and tf of the track; J), efficiency (ratio between the displacement and the distance covered by the whole track; K), net velocity (μm/min, ratio between the displacement and total time of the track; L) in control and inhibitor treated explant cultures. Dots, individual cells tracked (from n = 2 experiments containing control and inhibitor treatments). Mean ± SEM. Statistical significance was assessed by a Mann-Whitney test. pc, pharyngeal constrictor. Scale bar: D, 100 μm.

DOI: https://doi.org/10.7554/eLife.47460.009

The following video, source data, and figure supplement are available for figure 4:

**Source data 1.** This. zip file contains excel tables with the individual tracking parameters.
DOI: https://doi.org/10.7554/eLife.47460.011
**Figure supplement 1.** *Met* invalidation does not affect the proliferation and survival of *Isl1* progenitors.
DOI: https://doi.org/10.7554/eLife.47460.010
**Figure 4—video 1.** Time-lapse movie of a control E14.5 *Isl1^{Cre};R26^{mT/mG}* esophagus explant culture.
DOI: https://doi.org/10.7554/eLife.47460.012
**Figure 4—video 2.** Time-lapse movie of a E14.5 *Isl1^{Cre};R26^{mT/mG}* esophagus explant culture treated with 10 μM PF-0417903.
DOI: https://doi.org/10.7554/eLife.47460.013
**Figure 4—video 3.** Time-lapse movie of a E14.5 *Isl1^{Cre};R26^{mT/mG}* esophagus explant culture treated with 10 μM MGCD-265.
DOI: https://doi.org/10.7554/eLife.47460.014

esophagus are the major source of *Hgf* for anterior-posterior migration of myogenic progenitors. As such, this *Hgf* dynamic pattern might contribute to myogenic cell progression from the outer to inner layer, and anterior to posterior levels.

In turn, *Met* was expressed at high levels anteriorly in *Isl1*-derived mGFP+ myogenic cells, and also in luminal epithelial cells (**Figure 5C**, **Figure 5—figure supplement 1G,H**). However, the levels of *Met* transcript in mGFP+ cells were heterogeneous. Co-immunostaining with Myod and Myog antibodies to detect differentiating myogenic cells, revealed that 34% of mGFP+ cells were Myod+/Myog+ and that 74% of these Myod+/Myog+ cells had low levels of *Met* transcript (score 0 and 1; **Figure 5C1,C2,D,E**). Conversely, 91% of the Myod-/Myog- cells, expressed high levels of *Met* (score 3 and 4) (**Figure 5C1,C2,E**). Therefore, expression of *Met* was inversely correlated with the differentiation status of *Isl1*-derived cells.

To investigate the relative expression status of these myogenic markers in more detail, we performed single cell RT-qPCR analysis of *Isl1*-derived ESM progenitors. The mononucleated cell front in the esophagus of E15.5 *Isl1^{Cre};R26^{mT/mG}* mice was dissected and the expression of myogenic markers was examined in mGFP+ isolated by fluorescence-activated cell sorting (FACS) (**Figure 5—figure supplement 2A**). The normalized relative expression of the studied genes for all the filtered cells was annotated in a heatmap (**Figure 5—figure supplement 2B**) and revealed two groups of genes. The first group included *Isl1*, *Met*, *Pax7* and *Myf5* which were detected in nearly all the single cells analyzed. The second group included *Mrf4*, *Myog* and *Myod* which were expressed in a subset of cells. To assess the degree of relatedness between genes, we calculated the Spearman's correlation between all pairs of genes and noted that the expression of *Isl1*, *Met*, *Pax7* and *Myf5* was significantly positively correlated (**Figure 5F**). In contrast, *Isl1* expression showed no significant correlation with the expression of more downstream MRF genes (Myod, Myog, and Mrf4). This indicates that *Isl1* is associated with the upstream state, and that *Isl1* and *Met* likely act concomitantly in myogenic progenitors during ESM formation.

## Discussion

In vertebrates, branchiomeric head and neck muscles share a common CPM progenitor pool regulated by upstream molecular players including *Tbx1* and *Isl1*. Here, we uncover a cell-autonomous requirement of *Tbx1* and *Isl1* in the specification of CPM-derived esophagus myogenic progenitors. In addition, we show for the first time a unique dependency of myogenic progenitors on MET/HGF signaling pathway for esophagus spatio-temporal patterning. Surprisingly, laryngeal muscles that

also originate from the posterior pharyngeal arches are unaffected in *Met* mutants, thereby uncoupling the genetic requirements between ontogenically similar groups of CPM-derived muscles. These findings highlight distinct genetic hierarchies operating with CPM derivatives, and provide a framework to address myopathies of branchiomeric origin (*Figure 6*).

## Cell-autonomous role of *Isl1* during esophagus myogenesis

Recent genetic studies revealed that neck muscles including pharyngeal and laryngeal muscles, the trapezius and the esophagus originate from posterior pharyngeal arch mesoderm derived from an *Isl1*-lineage (*Gopalakrishnan et al., 2015*; *Heude et al., 2018*; *Lescroart et al., 2015*; *Tabler et al., 2017*). We previously demonstrated that *Tbx1* and *Isl1* genes play key upstream roles during ESM formation (*Gopalakrishnan et al., 2015*). In *Tbx1*-null embryos, *Isl1*-derived ESM fail to form, indicating that *Tbx1* acts upstream of *Isl1* during esophagus myogenesis (*Gopalakrishnan et al., 2015*). Given that Isl1 promotes cell proliferation and represses myogenic differentiation, Isl1 has been proposed to exert a conserved role in the specification of CPM progenitors (*Cai et al., 2003*; *Diogo et al., 2015*; *Harel et al., 2009*). However, the intrinsic role of Isl1 in CPM derivatives has not been addressed due to early embryonic lethality (*Cai et al., 2003*). Here, by means of chimeric analysis, we show that *Isl1*-null ES cells are specifically excluded from the ESM indicating that *Isl1* acts cell-autonomously during ESM formation at fetal stages, further supporting its role in the specification of branchiomeric myogenic progenitors.

## MET/HGF signaling drives anterio-posterior migration of esophagus muscle progenitors

During development, positional information that includes migration cues is often imparted to cells through intercellular signaling to allow proper spatio-temporal patterning. Several studies have uncovered the role of MET receptor and its ligand HGF in the proliferation and long-range migration of myogenic progenitors (*Trusolino et al., 2010*). The *Met* allele used here (*Met^D*) carries a mutation in two phosphotyrosines (Tyr1349, Tyr1356) in the carboxy-terminal tail, which completely abrogates MET function and recapitulates the *Met* null phenotype (*Bladt et al., 1995*; *Maina et al., 1996*; *Maina et al., 2001*). Previous work showed that in *Met* and *Hgf* mutants, *Pax3*-derived hypaxial muscles are missing, while other epaxial trunk muscle groups appear unaffected (*Bladt et al., 1995*; *Dietrich et al., 1999*; *Maina et al., 1996*; *Prunotto et al., 2004*).

In the trunk, *Hgf* is first expressed adjacent to somites, and subsequently along the migratory route and at target sites in limb connective tissue (*Dietrich et al., 1999*). In the esophagus, we identified the smooth muscle layer, which serves as a scaffold for myogenic progenitor migration, to be a major source of *Hgf*. This finding is in agreement with a recent study showing that HGF is mainly localized in smooth muscle cells in endodermal organs including the stomach and esophagus (*Jangphattananont et al., 2019*). Thus, our data highlight the functional coordination between adjacent but distinct smooth and striated muscle progenitors that facilitate ESM patterning which is unique compared to muscle patterning elsewhere.

How HGF levels are precisely controlled in the ESM developmental context to allow A-P muscle progenitor cell migration is unknown. In the limb, *Hgf* transcripts retreat in the subectodermal region, but a proximo-distal *Hgf* transcript gradient along the migration route of Met+ myogenic progenitors is not clearly observed (*Birchmeier and Gherardi, 1998*; *Bladt et al., 1995*; *Dietrich et al., 1999*; *Yang et al., 1996*). In the ESM, we observed by in situ hybridization on whole mount and sections that *Hgf* is expressed throughout the length of the esophagus at the time of seeding (E13.5) and ongoing migration (E14.5, E16.5) in a bilayered pattern. Unexpectedly, we observed seemingly constant transcript levels between the migratory front and more posterior levels raising the question of how directed migration is promoted. Interestingly, we observed by qRT-PCR and in situ hybridization that *Hgf* levels are diminished anteriorly in the outer layer at fetal stages, upstream to the front, concomitant with a decrease in *SMA* transcript and protein levels. Thus, it is possible that a gradient for migration is established by a decrease of *Hgf* source anteriorly, given the decrease in size and number of smooth muscle cells that occurs cranially from fetal stages (*Zhao and Dhoot, 2000a*). The fate of smooth muscle cells has been debated, nevertheless a combination of cell loss and distal compaction of smooth muscle cells, appear to be contributing factors (*Krauss et al., 2016*; *Rishniw et al., 2007*). Thus, the observed *Hgf* dynamics with respect to the

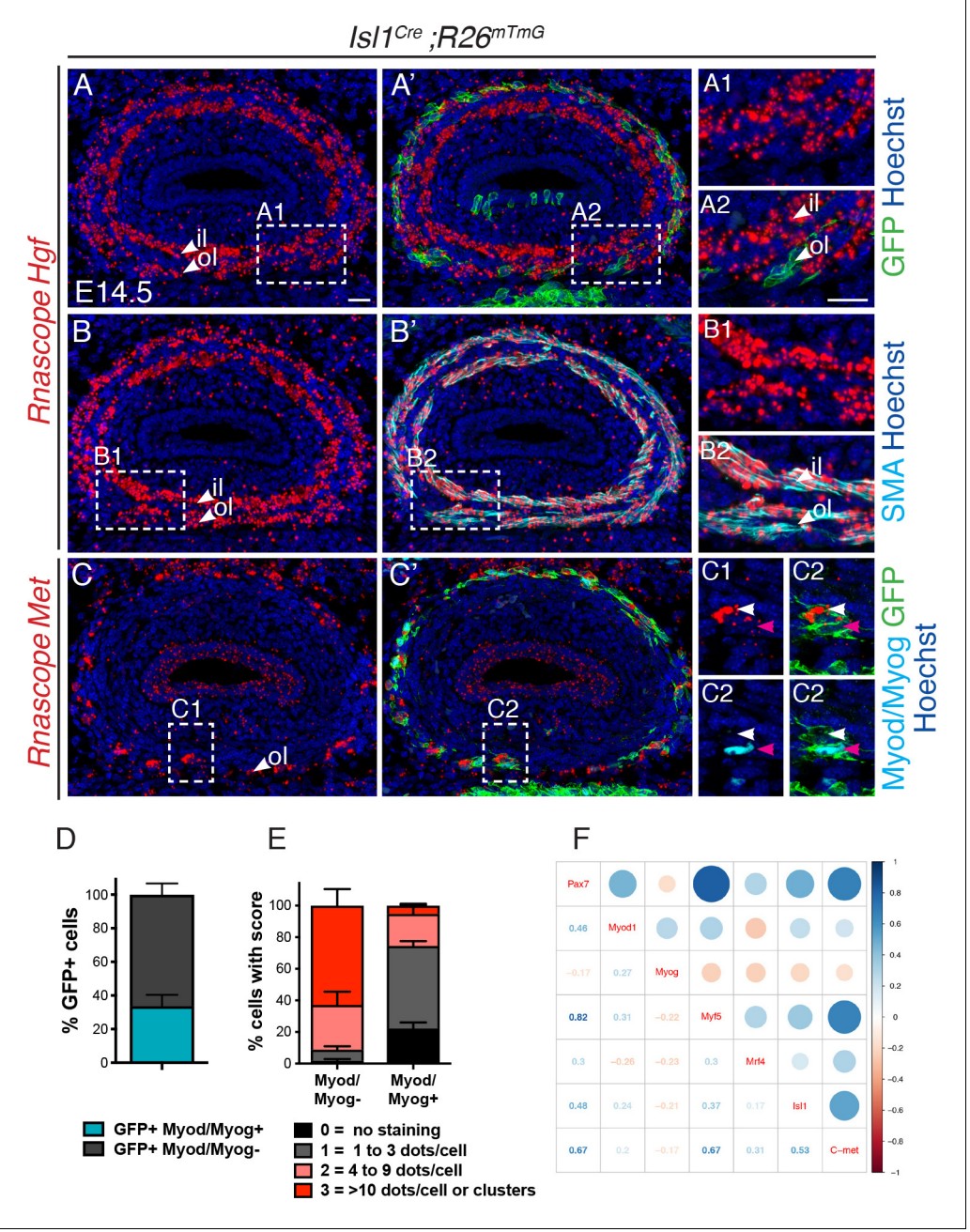

**Figure 5.** *Met* is expressed in undifferentiated *Isl1*-derived myogenic progenitors. (**A–C**) In situ hybridization on transverse cryosections at the esophagus level of E14.5 *Isl1^{Cre};R26^{mTmG}* embryos for *Hgf* (**A, B**) and *Met* (**C**), combined with immunofluorescence for GFP (Isl1-derived progenitors), SMA (smooth muscle actin) and Myod/Myog (myogenic cells) (shown in A'-C'). Note that *Hgf* is expressed adjacent to Isl1-derived cells (**A1,A2**) by SMA+ cells (**B1,B2**). *Met* is expressed by Isl1-derived cells but at levels inversely correlated to Myod/Myog+ expression (C1, C2, channels split for clarity). Note that *Isl1*-derived myogenic cells are exclusively present in the outer layer at this stage. (**D**) Histograms of the percentage of Myod/Myod- and Myod/Myog+ Isl1-derived GFP+ cells. (**E**) Histograms of the percentage of cells in (**D**) with a defined RNAscope score for *Met* expression. n = 3 embryos, with a minimum of 3 sections at the anteriormost part of the esophagus per embryo. A total of 368 GFP+ cells were assessed for the expression of Myod/Myog (**D**) and RNAscope score. (**F**) Correlogram. The upper part of the mixed correlogram displays graphically the degree of relationships between genes. The bigger the circle, the higher the Spearman's correlation coefficient; the redder, the more negative; the bluer, the more positive. The lower part shows the values of the Spearman's coefficient. il, inner layer ; ol, outer layer. Scale bars: A, A2, 20 µm.
DOI: https://doi.org/10.7554/eLife.47460.015

*Figure 5 continued on next page*

*Figure 5 continued*

The following source data and figure supplements are available for figure 5:

**Source data 1.** Excel table containing individual countings from three independent experiments to generate the histograms in panels 5D-E.

DOI: https://doi.org/10.7554/eLife.47460.018

**Source data 2.** This folder contains the initial single cell count matrix of the Ct values (tab2_3_sc_edge_allGenes_Ct.R), and the R source code (eso_t2_t3__analysis.R) used to filter, log-transformed, normalized, calculate the correlation coefficient and generate the correlogram on this figure.

DOI: https://doi.org/10.7554/eLife.47460.019

**Figure supplement 1.** Expression domains of *Hgf* and *Met* along the developing esophagus.

DOI: https://doi.org/10.7554/eLife.47460.016

**Figure supplement 2.** Esophagus single cell analysis.

DOI: https://doi.org/10.7554/eLife.47460.017

---

localization of myogenic cells are in agreement with the outer to inner layer and anterior to posterior myogenic cell progression (*Zhao and Dhoot, 2000a*; *Zhao and Dhoot, 2000b*).

In addition to transcriptional control, directional migration may also rely on a gradient of active two-chain HGF heterodimer. HGF bioavailability depends on a number of factors including HGF activators and inhibitors that exert essential roles during embryonic development and muscle regeneration (*Rodgers et al., 2017*; *Sisson et al., 2009*; *Tanaka et al., 2005*; *van Adelsberg et al., 2001*) and heparin sulfate proteoglycans that can enhance MET/HGF signaling (*Gutiérrez et al., 2014*). In addition, myogenic cells at the migratory front might sequester active HGF, thereby limiting its bioavailability. As such, net caudal movement could result from a self-generated localized signaling gradient as has been observed in other developmental contexts (*Cai and Montell, 2014*).

In our time-lapse movies, we noted that *Isl1*-derived progenitors navigate the esophageal scaffold, switching directions, but with a net caudal displacement as differentiated myofibers are deposited in its wake. Thus, another possibility is that HGF does not act as a directional migration cue, but rather maintains ESM myogenic progenitors in a scattered status to reach a 'myogenic free' zone. The functional validation for these diverse scenarios awaits further investigation.

### *Met* regulation underlies esophagus myogenic patterning

It has been proposed that a prolonged interaction between MET and HGF may be required to prevent cell re-aggregation, thereby maintaining cell motility and preventing expression of the MRFs (*Dietrich et al., 1999*). The first obvious deficiency observed in $Met^{D/D}$ embryos is seen at the time Isl1+ progenitors colonize the smooth muscle scaffold (by E14.5). In controls, colonization progresses posteriorly and myofibers are formed while maintaining a pool of progenitor cells. In the mutant, only few *Isl1*-derived myofibers are present in the upper esophagus. Similarly, our pharmacological inhibition studies of MET receptor activity in esophagus explants resulted in impaired progenitor cell migration and precocious differentiation. Thus, MET/HGF signaling might have a role in maintenance of the undifferentiated state of migratory muscle progenitors to allow continuous progression of the myogenic front. It remains unclear if motility prevents or delays expression of the differentiation genes, or if expression of differentiation genes stops motility. However, it has been shown that precocious expression of MRFs in dermomyotomal muscle progenitors prevents their migration into limb buds (*Bonnet et al., 2010*), while application of HGF results in reduction of *Myod* expression (*Scaal et al., 1999*).

During limb muscle development, the *Met* receptor was reported to be under the direct transcriptional regulation of *Pax3* (*Epstein et al., 1996*). Interestingly, ESM development requires MET/HGF signaling in a Pax3-independent context. Hence the upstream modulator of *Met* expression in ESM progenitors remains an open question. *Pax3* and its paralog *Pax7* have partially redundant functions in muscle progenitors (*Relaix et al., 2006*). Given that common Pax3/Pax7 binding sites are found in *Met* regulatory regions in adult limb primary myoblasts (*Soleimani et al., 2012*), Pax7 could exert such a role in ESM progenitors. Interestingly, *Pax7* knockout mice develop megaesophagus postnatally with striated muscle present in an abnormally proximal position (*Chihara et al., 2015*). Whether this impairment in ESM formation is solely due to reduced proliferation and precocious differentiation at the migratory front, or is concomitant to a reduced expression of MET is currently

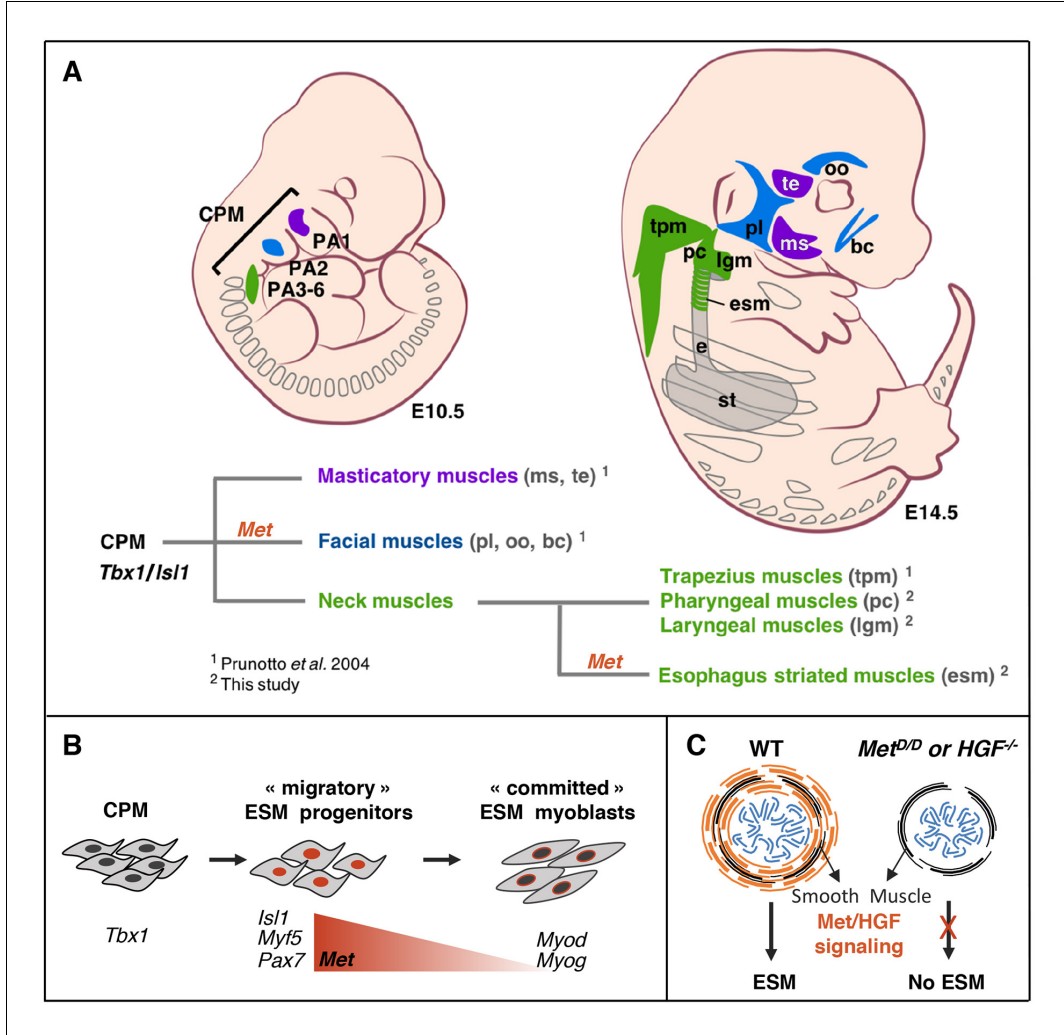

**Figure 6.** *Tbx1-Isl1-Met* genetic pathway regulates only a subset of CPM-derived muscles. (**A**) Masticatory (purple) and facial (blue) muscles originating from anterior pharyngeal arches (PA1-2) are indicated. Neck muscles (green) derived from posterior PAs including trapezius, pharyngeal and laryngeal muscles, develop in a *Met*-independent context, while esophagus striated muscles are under the control of MET/HGF signaling. (**B**) A *Tbx1/Islet1/Met* genetic hierarchy acts in uncommitted ESM progenitors. Then, *Met* expression decreases in myoblasts during myogenic commitment. (**C**) Absence of ESM formation in the *Met* and *Hgf* mutants. CPM, cardiopharyngeal mesoderm; bc, buccinator; e, esophagus; esm, esophagus striated muscles; lgm, laryngeal muscles; ms, masseter; oo, orbicularis oculi; PA1-6, pharyngeal arches 1–6; pc, pharyngeal constrictor; pl, platysma; st, stomach; te, temporal; tpm, trapezius muscles.

DOI: https://doi.org/10.7554/eLife.47460.020

unknown. On the other hand, our in situ hybridization and single cell qPCR data showed that *Isl1*, *Pax7* and *Met* are predominantly co-expressed in uncommitted ESM progenitors (Pax7+, Myf5+, Myod/Myog-), with decreased expression in committed cells. This observation is in agreement with the known role of Isl1 as negative regulator of muscle differentiation of CPM-derived muscles (*Harel et al., 2009*). It remains unclear if Isl1 also exerts a direct role in motility by regulation of *Met*. Interestingly, Isl1-Lhx3 fusion protein was shown to induce *Met* expression in motor neurons (*Lee et al., 2012*). Intriguingly, several putative Lhx binding sites including consensus Isl1-binding sites (cTAATg) were identified by in silico analysis of *Met* promoter elements using matinspector (*Cartharius et al., 2005*) (data not shown). It is therefore tempting to speculate a direct regulation of *Met* expression by Isl1.

Finally, a hierarchy in lineage progression could be inferred from phenotypes of mutant mice (*Krauss et al., 2016*). The myogenic front in the *Pax7* knockout esophagus is localized more posteriorly than what we observed in *Met*-null embryos (*Chihara et al., 2015*). This suggests a possible temporal regulation of *Met* expression in ESM progenitors by Isl1 in the fetus, preempted by Pax7 in postnatal stages, and likely facilitated by temporally controlled transcriptional coactivators. Taken together, these observations warrant further studies including genome-wide ChIP-seq to explore if Pax7 and/or Isl1 directly regulate *Met* expression.

## Myogenic diversity within CPM-derived muscles

An unexpected finding from our work is that CPM muscles originating from posterior pharyngeal arches are differentially affected in *Met* mutants. Head muscles derived from the second pharyngeal arch and giving rise to branchiomeric facial muscles (orbicularis oculi, buccinator, platysma) appear either strongly reduced or absent, while first arch-derived masticatory (masseter, temporalis) and extraocular muscles are present in $Met^{D/D}$ mutants (*Prunotto et al., 2004*). However, we show that posterior branchiomeric neck muscles, including pharyngeal and laryngeal muscles, are present in $Met^{D/D}$ mutants while adjacent ESM is absent. Thus, we have established a unique *Tbx1-Isl1-Met* genetic hierarchy in ESM progenitors that is distinct from other posterior branchiomeric muscles (*Figure 6*).

The genetic regulatory pathways that give rise to functionally distinct groups of muscles has provided critical information to understand heterogeneity in response to genetic diseases, such as DiGeorge syndrome where mutations in TBX1 result in the impaired function of subsets of craniofacial and pharyngeal apparatus with varied degrees of severity. Understanding the functional dynamics of *Tbx1* and *Isl1* in specific muscles groups will help uncover differences between the ontogenically similar subsets of CPM-derived muscles. Uncoupling the genetic requirements of these distinct populations is necessary to provide a framework that will explain how human myopathies affect only subsets of muscles (*Emery, 2002*; *Randolph and Pavlath, 2015*).

# Materials and methods

**Key resources table**

| Reagent type (species) or resource | Designation | Source or reference | Identifiers | Additional information |
|---|---|---|---|---|
| Strain, strain background (*Mus musculus*) | B6D2F1/JRj | Janvier | | |
| Genetic reagent (*M. musculus*) | $Islet1^{Cre}$ | PMID:11299042 | MGI:2447758 | Dr. Thomas M Jessell (Howard Hughes Medical Institute, Columbia University, USA) |
| Genetic reagent (*M. musculus*) | $Myf5^{nlacZ}$ | PMID:8918877 | MGI:1857973 | Dr. Shahragim Tajbakhsh (Department of Developmental and Stem Cell Biology, Institut Pasteur, France) |
| Genetic reagent (*M. musculus*) | Tg :Pax7-nGFP | PMID:19531352 | MGI:5308730 | Dr. Shahragim Tajbakhsh (Department of Developmental and Stem Cell Biology, Institut Pasteur, France) |
| Genetic reagent (*M. musculus*) | $R26^{mT/mG}$ | PMID:17868096 | MGI:3716464 | Pr. Philippe Soriano (Icahn School of Medicine at Mt. Sinai, USA) |
| Genetic reagent (*M. musculus*) | Hgf KO | PMID:7854452 | MGI:1857656 | Pr. Carmen Birchmeier (Max Delbruck Center for Molecular Medicine, Germany) |

*Continued on next page*

*Continued*

| Reagent type (species) or resource | Designation | Source or reference | Identifiers | Additional information |
|---|---|---|---|---|
| Genetic reagent (*M. musculus*) | *Met$^D$* | PMID:8898205 | MGI:1858019 | Pr. Carola Ponzetto (Department of Molecular Biotechnology, University of Turin, Italy) |
| Genetic reagent (*M. musculus*) | *Tbx1KO* | PMID:11242110 | MGI:2179190 | Dr. Virginia Papaioannou (Department of Genetics and Development, Columbia University Medical Center, USA) |
| Antibody | Chicken polyclonal anti-β-gal | Abcam | Cat. #: ab9361 RRID:AB_307210 | IF (1:1000) |
| Antibody | Rabbit polyclonal anti-β-gal | MP Biomedicals | Cat. #: MP 559761 RRID:AB_2687418 | IF (1:1500) |
| Antibody | Chicken polyclonal anti-GFP | Aves Labs | Cat. #: 1020 RRID:AB_10000240 | IF (1:500) |
| Antibody | Chicken polyclonal anti-GFP | Abcam | Cat. #: 13970 RRID:AB_300798 | IF (1:1000) |
| Antibody | Mouse monoclonal IgG1 anti-Islet1 | DSHB | Cat. #: 40.2D6 RRID:AB_528315 | IF (1:1000) |
| Antibody | Mouse monoclonal IgG1 anti-Desmin | Dako | Cat. #: ab8470 RRID:AB_306577 | IF (1:100) |
| Antibody | Mouse monoclonal IgG1 anti-Myod | Dako | Cat. #: M3512 RRID:AB_2148874 | IF (1:100) |
| Antibody | Mouse monoclonal IgG1 anti-Myod | BD-Biosciences | Cat. #: 554130 RRID:AB_395255 | IF (1:500) |
| Antibody | Mouse monoclonal IgG1 anti-Pax7 | DSHB | Cat. #: Pax7 RRID:AB_528428 | IF (1:20) |
| Antibody | Mouse monoclonal IgG2a anti-E-Cad | BD Biosciences | Cat. #: 610182 RRID:AB_397581 | IF (1:500) |
| Antibody | Mouse monoclonal IgG1 anti-Myog | DSHB | Cat. #: F5D RRID:AB_2146602 | IF (1:20) |
| Antibody | Rabbit polyclonal anti-SMA | Abcam | Cat. #: ab5694 RRID:AB_2223021 | IF (1:1000) |
| Antibody | Mouse monoclonal IgG1 anti-Tnnt3 | Sigma Aldrich | Cat. #: T6277 RRID:AB_261723 | IF (1:200) |
| Antibody | Mouse monoclonal IgG2a anti-Tuj1 | Ozyme/BioLegend | Cat. #: BLE801202 RRID:AB_2313773 | IF (1:1000) |
| Antibody | Alexa Fluor 633 F(ab')2 Fragment of Goat Anti-Rabbit IgG (H+L) | Life Technologies | Cat. #: A-21072 RRID:AB_2535733 | IF (1:500) |
| Antibody | Alexa Fluor 555 F(ab')2 Fragment of Goat Anti-Rabbit IgG (H+L) | Life Technologies | Cat. #: A-21430 RRID:AB_2535851 | IF (1:500) |
| Antibody | Alexa Fluor 488 F(ab')2 Fragment of Goat Anti-Rabbit IgG (H+L) | Life Technologies | Cat. #: A-11070 RRID:AB_2534114 | IF (1:500) |
| Antibody | Alexa Fluor 633 Goat Anti-Chicken IgG (H+L) | Life Technologies | Cat. #: A-21103 RRID:AB_2535756 | IF (1:500) |
| Antibody | Alexa Fluor 488 Goat Anti-Chicken IgG (H+L) | Life Technologies | Cat. #: A-11039 RRID:AB_2534096 | IF (1:500) |

*Continued on next page*

*Continued*

| Reagent type (species) or resource | Designation | Source or reference | Identifiers | Additional information |
|---|---|---|---|---|
| Antibody | Alexa Fluor 633 Goat Anti-Mouse IgG1 (γ1) | Life Technologies | Cat. #: A 21126 RRID:AB_2535768 | IF (1:500) |
| Antibody | Alexa Fluor488 AffiniPure Goat Anti-Mouse IgG1 (γ1) | Jackson ImmunoResearch | Cat. #: 115-545-205 RRID:AB_2338854 | IF (1:500) |
| Antibody | Cy3-AffiniPure Goat Anti-Mouse IgG1 (γ1) | Jackson ImmunoResearch | Cat. #: 115-165-205 RRID:AB_2338694 | IF (1:500) |
| Antibody | Cy3-AffiniPure Goat Anti-Mouse IgG2a (γ2a) | Jackson ImmunoResearch | Cat. #: 115-165-206 RRID:AB_2338695 | IF (1:500) |
| Antibody | Dylight 405 Goat Anti-Mouse IgG2a (γ2a) | Jackson ImmunoResearch | Cat. #: 115-475-206 RRID:AB_2338800 | IF (1:500) |
| Commercial assay, kit | RNAscope 2.5 HD reagent Kit-RED | ACD/Bio-techne | Cat. #: 322350 | |
| Commercial assay, kit | RNAscope Multiplex Fluorescent reagent Kit-V2 | ACD/Bio-techne | Cat. #: 323100 | |
| Commercial assay, kit | RNAscope Probe – Mm-Hgf (C1) | ACD/Bio-techne | Cat. #: 315631 | |
| Commercial assay, kit | RNAscope Probe – Mm-Met (C1) | ACD/Bio-techne | Cat. #: 405301 | |
| Commercial assay, kit | Opal 570 Reagent Pack | PerkinElmer | Cat. #: FP1488001KT | 1 :1500 of reconstituted reagent in RNAscope Multiplex TSA Buffer |
| Sequence-based reagent | qPCR Primer TBP Fw | This paper | ATCCCAAGCGATTTGCTG | Materials and methods, Quantitative RT-qPCR section |
| Sequence-based reagent | qPCR Primer TBP Rev | This paper | CCTGTGCACACCATTTTTCC | Materials and methods, Quantitative RT-qPCR section |
| Sequence-based reagent | qPCR Primer Isl1 Fw | *Gopalakrishnan et al., 2015* | CGTGCTTTGTTAGGGATGGGA | |
| Sequence-based reagent | qPCR Primer Isl1 Rev | *Gopalakrishnan et al., 2015* | AGTCGTTCTTGCTGAAGCCT | |
| Sequence-based reagent | qPCR Primer Myf5 Fw | This paper | GACAGGGCTGTTACATTCAGG | Materials and methods, Quantitative RT-qPCR section |
| Sequence-based reagent | qPCR Primer Myf5 Rev | This paper | TGAGGGAACAGGTGGAGAAC | Materials and methods, Quantitative RT-qPCR section |
| Sequence-based reagent | qPCR Primer Met Fw | *Sambasivan et al., 2009* | GCATTTTTACGGACCCAACC | |
| Sequence-based reagent | qPCR Primer Met Rev | *Sambasivan et al., 2009* | TTCACAGCCGGAAGAGTTTC | |
| Sequence-based reagent | qPCR Primer Hgf Fw | This paper | CTTCTCCTTGGCCTTGAATG | Materials and methods - Quantitative RT-qPCR section |
| Sequence-based reagent | qPCR Primer Hgf Rev | This paper | AGGCCATGGTGCTACACTCT | Materials and methods - Quantitative RT-qPCR section |
| Sequence-based reagent | qPCR Primer SMA Fw | This paper | CTCTCTTCCAGCCATCTTTCAT | Materials and methods - Quantitative RT-qPCR section |
| Sequence-based reagent | qPCR Primer SMA Rev | This paper | TATAGGTGGTTTCGTGGATGC | Materials and methods - Quantitative RT-qPCR section |
| Sequence-based reagent | Taqman qPCR Primers Tbp | ThermoFisher Scientific | Cat. #: Mm00446971_m1 | Sequence not available, probe spans exons |
| Sequence-based reagent | Taqman qPCR Primers Actb | ThermoFisher Scientific | Cat. #: Mm00607939_s1 | Sequence not available, primers and probe map within a single exon |
| Sequence-based reagent | Taqman qPCR Primers Hprt | ThermoFisher Scientific | Cat. #: Mm01545399_m1 | Sequence not available, probe spans exons |

*Continued on next page*

*Continued*

| Reagent type (species) or resource | Designation | Source or reference | Identifiers | Additional information |
|---|---|---|---|---|
| Sequence-based reagent | Taqman qPCR Primers Rpl13a | ThermoFisher Scientific | Cat. #: Mm01612987_g1 | Sequence not available, probe spans exons |
| Sequence-based reagent | Taqman qPCR Primers Rps29 | ThermoFisher Scientific | Cat. #: Mm02342448_gH | Sequence not available, probe spans exons |
| Sequence-based reagent | Taqman qPCR Primers Pax7 | ThermoFisher Scientific | Cat. #: Mm01354484_m1 | Sequence not available, probe spans exons |
| Sequence-based reagent | Taqman qPCR Primers Myod | ThermoFisher Scientific | Cat. #: Mm01203489_g1 | Sequence not available, probe spans exons |
| Sequence-based reagent | Taqman qPCR Primers Myog | ThermoFisher Scientific | Cat. #: Mm00446195_g1 | Sequence not available, probe spans exons |
| Sequence-based reagent | Taqman qPCR Primers Pax3 | ThermoFisher Scientific | Cat. #: Mm00435491_m1 | Sequence not available, probe spans exons |
| Sequence-based reagent | Taqman qPCR Primers Myf5 | ThermoFisher Scientific | Cat. #: Mm00435125_m1 | Sequence not available, probe spans exons |
| Sequence-based reagent | Taqman qPCR Primers Mrf4 | ThermoFisher Scientific | Cat. #: Mm00435127_g1 | Sequence not available, probe spans exons |
| Sequence-based reagent | Taqman qPCR Primers Isl1 | ThermoFisher Scientific | Cat. #: Mm00517585_m1 | Sequence not available, probe spans exons |
| Sequence-based reagent | Taqman qPCR Primers Met | ThermoFisher Scientific | Cat. #: Mm00436382_m1 | Sequence not available, probe spans exons |

## Animals

Animals were handled as per European Community guidelines and the ethics committee of the Institut Pasteur (CTEA) approved protocols. $Isl1^{Cre}$ (*Srinivas et al., 2001*), reporter mouse lines $R26R^{mT/mG}$ (*Muzumdar et al., 2007*), $Myf5^{nLacZ}$ (*Tajbakhsh et al., 1996*), *Tg: Pax7nGFP* (*Sambasivan et al., 2009*), and mutant mice carrying the $Tbx1^{tm1pa}$ allele (referred to as $Tbx1^{-/-}$) (*Jerome and Papaioannou, 2001*), *Hgf* (*Schmidt et al., 1995*) and *Met* (referred as $Met^D$) (*Maina et al., 1996*) mutant alleles were described previously. To generate experimental embryos for $Met^{D/D}$ together with *Isl1* and *Myf5* lineage tracings, $Met^{D/+}: Isl1^{Cre/+}: Myf5^{nlacZ/+}$ males were crossed with $Met^{D/+}: R26R^{mTmG/mTmG}$ females. Mice were kept on a mixed genetic background C57BL/6JRj and DBA/2JRj (Janvier Labs). Mouse embryos and fetuses were collected between embryonic day (E) E12.5 and E18.5, with noon on the day of the vaginal plug considered as E0.5.

## Generation of *Isl1*-null chimeras

For derivation of *Isl1-null* ES cells, males and females from $Isl1^{Cre/+}$ genotype (*Srinivas et al., 2001*) were intercrossed to produce heterozygous and homozygous *Isl1-null* blastocysts. At E3.5, blastocysts were collected from uterine horns and put on culture for 3–6 days in ES derivation medium composed of GlutaMAX/DMEM (Gibco, 31966), 15% FBS (Biowest), 1% penicillin/streptomycin (Gibco, 15140, stock 100X), 1% sodium pyruvate (Gibco, 11360, stock 100 mM), 0.1% β-mercaptoethanol (Gibco, 31350–010, stock 50 mM), 1000 U/ml ESGRO recombinant mice leukemia inhibitory factor (LIF, Millipore, ESG1107, stock $10^7$ U/ml) and 2i (1 µM PD325; Axon Medchem 1408; and 1 µM CH99; Axon Medchem 1386) on gelatin-coated wells with primary Mouse Embryonic Fibroblasts (MEFs). The disaggregation of ICM was performed with 5 min of 0.05% trypsin-EDTA (GIBCO, 25300–054) treatment and the cell suspension put on culture in ES derivation medium on MEFs. Derived ES cells were then expanded and genotyped by PCR with specific primers for amplification of *Isl1* WT and mutant sequences (WT primers: ccaagtgcagcataggcttcag; gcagaggccgcgctggatgcaagg, 230 bp; Mutant primers: tcatgcaagctggtggctgg; gcagaggccgcgctggatgcaagg, 633 bp).

To trace the ES clones, CAG-nlacZ and PGK-puro cassettes were cloned into a pBluescript to produce a nlacZ reporter puromycin resistant plasmid. Heterozygous and homozygous *Isl1-null;nLacZ* ES cells were electroporated (0.5–1 × $10^7$ cells) with 20 µg of linearized pCAGnlacZ-puro plasmid by using a BTX Harvard apparatus ECM830 electroporator with one pulse at 240V for 15 ms. Three days after transfection, positive clones were selected in ES derivation medium with puromycin (1.5

µg/ml) for 5 days. ES colonies were picked into 24-well plates and tested for expression of the *nLacZ* reporter (X-Gal/immunostaining).

For chimera production, the β-gal+ selected clones were further expanded in ES culture medium (ES derivation medium without 2i) on MEFs. C57BL/6N females were superovulated and mated with C57BL/6N males. At E3.5, blastocysts were collected, injected with wildtype (control) or homozygous *Isl1*-null;*nlacZ* ES cells (2–6 cells/blastocyst) and were subsequently transferred into the uterus of 0.5 or 2.5 dpc pseudopregnant B6CBAF1 females (15–17 blastocysts/females). Chimeric fetuses were harvested at E16.5 or E18.5 for analysis. The collected fetuses were dissected in PBS at 4°C to remove the caudal part below the stomach, then fixed 3 hr at 4°C in 4% paraformaldehyde (PFA, Electron Microscopy Sciences, 15710), 0.5% Triton X-100 (SIGMA, T8787) and extensively washed in PBS with 0.1% Tween 20 (PBST) (SIGMA, P1379). To evaluate the contribution of ES cells to specimens, X-gal staining was performed on the dissected lower part of the fetuses. The analysis was performed by immunofluorescent stainings on cryosections of the rostral part of the fetuses.

## Generation of *Tbx1*-null chimeras

The Isl1 nuclear lacZ (nlacZ) knock-in mouse 129/SV ES line (*Isl1$^{lacZ}$*) was obtained from Sylvia Evans (*Sun et al., 2007*). ES cells were cultured on Mytomycin-C treated embryonic primary fibroblasts onto gelatin coated dishes in DMEM-KO media (Gibco, 10829–018) containing 15% FBS (Biowest), 0,5% penicillin/streptomycin (Gibco, 15140, stock100x), 0.1% β-mercaptoethanol (SIGMA, M7522, stock 100 mM in PBS), 1% L-Glutamine (Gibco, 25030024, stock 200 mM) and 1000 U/ml ESGRO recombinant mice leukemia inhibitory factor (LIF, Millipore, ESG1107, stock 10$^7$ U/ml).

For ES cell injection and chimera production, 4-week-old *Tbx1$^{+/-}$* females were superovulated and mated with *Tbx1$^{+/-}$* males (on a mixed genetic background C57BL/6JRj and DBA/2JRj, Janvier Labs). At E3.5, blastocysts were collected, injected with *Isl1$^{lacZ}$* ES cells (6–12 cells/blastocyst) and were subsequently transferred into uteri of 0.5 or 2.5 dpc pseudopregnant B6CBAF1 females.

Chimeric fetuses were harvested at E14.5/E15.5 for analysis. The collected fetuses were fixed 2.5 hr at 4°C in 4% paraformaldehyde 0.2% Triton X-100 and extensively washed in PBS at 4°C. For genotyping of the chimeric embryos, the visceral yolk sac layers were separated using the trypsin/pancreatin method as described in *Wallingford and Giachelli (2014)* with modifications. Briefly, yolk sacs were collected and incubated in Ca$^{2+}$/Mg$^{2+}$-free Tyrode Ringer's saline solution containing 0.5% Trypsin (Gibco, 15090–046) and 2.5% Pancreatin (SIGMA, P-3292) for 4 hr at 4°C on individual wells of a 12 well plastic dish. Yolk sacs were then washed in GlutaMAX/DMEM (Gibco, 31966021) media buffered with 25 mM HEPES (SIGMA, H0887) and then transferred into media containing 10% FBS for at least 30 min at 4°C. The visceral endoderm (VEnd) and extraembryonic mesoderm (ExM) tissue layers of the visceral yolk sac were mechanically separated for genotyping. The VEnd layer is contributed exclusively by the host embryo while the ExM has dual contribution from ES cells and host embryo. DNA extraction was performed using Proteinase K and PCR performed with the following primers: Tbx1_for: tgcatgccaaatgtttccctg, Tbx1_rs: gatagtctaggctccagtcca, Tbx1_rs_Neo: agggccagctcattcctcccac (WT band: 196 bp; Mutant band: 450 bp), lacZ_fw: atcctctgcatggtcaggtc, lacZ_rs: cgtggcctgattcattcccc.

For the analysis of lacZ+ chimeric embryos, the digestive tract including the pharynx, trachea, esophagus, heart, stomach and diaphragm was further dissected and X-Gal stained overnight at 37°C or embryos were processed for cryosections and immunostaining on sucrose/OCT as described above.

## X-Gal staining and immunofluorescence

Wholemount samples were analyzed for β-galactosidase activity with 400 µg/ml X-Gal (SIGMA 15520–018; Stock solution 40 mg/ml in DMSO) in PBS buffer containing 4 mM potassium ferricyanide, 4 mM potassium ferrocyanide, 0.02% NP-40 and 2 mM MgCl$_2$ as previously described (*Comai et al., 2014*).

For immunostaining on cryosections, embryos and fetuses were fixed 3 hr in 4% PFA and 0,2–0,5% Triton X-100 at 4°C, washed overnight at 4°C in PBS, cryopreserved in 30% sucrose in PBS and embedded in OCT for cryosectioning. Cryosections (16–18 µm) were allowed to dry for 30 min and washed in PBS. For immunostaining on paraffin sections, samples were fixed overnight in 4% PFA, dehydrated in graded ethanol series, Histoclear II (HS-202, National Diagnostics) and embedded in

paraffin. Paraffin blocks were sectioned at 12 μm using a Leica microtome. Sections were then deparaffinised and rehydrated by successive immersions in Histoclear, ethanol and PBS series. When needed, samples were then subjected to antigen retrieval with 10 mM Citrate buffer (pH 6.0) using a 2100 Retriever (Aptum Biologics).

Rehydrated sections were blocked for 1 hr in 10% normal goat serum, 3% BSA, 0.5% Triton X-100 in PBS. Primary antibodies were diluted in blocking solution and incubated overnight at 4°C. After 3 rounds of 15 min washes in PBST, secondary antibodies were incubated in blocking solution 1 hr at RT together with 1 μg/ml Hoechst 33342 to visualize nuclei. Antibodies used in the study are listed in the Key Resource Table. After 3 rounds of 15 min washes in PBST, slides were mounted in 70% glycerol in PBS for analysis. For EdU staining, immunostaining for primary and secondary antibodies was performed first, followed by the click chemical reaction using Alexa633 as a reactive fluorophore for EdU detection (Life Technologies C10350).

For whole mount immunostaining, embryos were fixed and washed as above. Esophagi were micro-dissected in PBST and incubated in blocking buffer (10% goat serum, 10% BSA, 0.5% TritonX-100 in 1X PBS) for 1 hr at RT in 2 ml Eppendorff tubes. The tissue was then incubated with primary antibodies in the blocking buffer for 4–5 days at 4°C with rocking. The tissue was washed extensively for 2-4hr in PBST and then incubated in Fab' secondary antibodies for 2 days at 4°C with rocking. The tissue was washed as above, dehydrated in 50% Methanol in PBS, 100% Methanol and then cleared with BABB and mounted for imaging as in *Yokomizo et al. (2012)*.

## RNAscope in situ hybridization

E14.5 embryos were collected, fixed overnight in 4% PFA, washed in PBS 3 × 15 min, equilibrated in 15% and 30% sucrose and embedded in OCT. Tissue blocks were stored at −80C. 18 μm thick cryosections were collected on Superfrost Plus slides and stored at −80 till use (less than 2 months).

RNAscope probes Mm-*Hgf* (315631) and Mm-*Met* (405301) were designed commercially by the manufacturer and are available from Advanced Cell Diagnostics, Inc. In situ hybridization was performed using the RNAscope Multiplex Fluorescent Reagent Kit V2 and RNAscope 2.5 HD Reagent Kit-RED according to manufacturer's instructions (*Wang et al., 2012*) with some modifications. For sample pre-treatments: H2O2 treatment was 10 min at RT, retrieval was done for 2 min at 98°C and slides were digested with Protease Plus reagent for 15 min at 40°C. When the RNAscope 2.5 HD Reagent Kit-RED was used, the AMP1 to AMP6 steps were done as in the standard protocol. Before detection, samples were washed in PBS 3 × 5 min and immunostaining performed as above with fluorescent secondary antibodies. Sections were then washed in RNAscope Wash buffer, detection done with Fast-Red A/B mix and slides mounted in Fluoromount-G (InterBioTech, FP-483331). As the Fast-Red chromogenic precipitate is also visible by fluorescence microscopy using the 555 nm laser, sections were imaged using a 40x objective on a LSM700 microscope (Zeiss). When the RNAscope Multiplex Fluorescent V2 kit was used, detection of the probe was done with Opal570 reagent (Perkin Elmer, FP1488A, 1/1500 in TSA Buffer) prior to immunostaining. For quantitation of *Met* RNAscope staining, the number of individual signal dots or clusters per mGFP+ cell was counted manually on Fiji. Cells were attributed the score 1 (1 to 3 dots/cell), 2 (4 to 9 dots/cell) or 3 (more than 10 dots/cells or big clusters) and correlated to the presence or absence of Myod/Myog nuclear staining.

For quantitation of *Hgf* RNAscope staining per SMA area (*Figure 5—figure supplement 1F*), a manual ROI outlining the SMA+ layers was defined in Fiji. Segmentation of the channels was done using the defaut Auto Threshold for the *Hgf*/RNAscope channel and the Huang Auto Threshold for the SMA channel. The measure command was used to calculate the area of the ROI limited to the threshold for both channels.

## Enzymatic digestion for cell sorting

The masseter muscles and esophagi from *Tg:Pax7-nGFP* timed embryos were dissected in cold PBS and kept in cold GlutaMAX/DMEM (Gibco, 31966) with 1% Penicillin–Streptomycin. For single cell qPCR analysis, only mGFP+ cells from the mononucleated cell front (mcf) of the esophagus of *Isl1-Cre:R26mTmG* embryos were micro-dissected under a Zeiss SteREO Discovery V20 macroscope. Samples were processed with enzymatic digestion mix containing 0.1% Trypsin (15090–046,Gibco), 0,08% Collagenase D (Roche, 11088882001) and 10 μg/ml of DNAse I (04536282001, Roche) in

DMEM/Glutamax. Samples were incubated for 15 min at 37°C under 300 rpm agitation and resuspended by gently pipetting up and down 10–15 times using a P1000 pipette. Incubation and resuspension by pipetting were repeated for two additional 15 min enzymatic treatments. The digests were passed through a 70 micron then 40 micron SmartCell Strainers (Milteny Biotec) and digestion was stopped with fetal bovine serum (FBS, Gibco). Cells were spun at 600 g 15 min at 4°C and the pellets resuspended in 300 µl of DMEM/2% FBS to be processed for FACS.

## Quantitative RT-qPCR

Total RNA from esophagus portions and limbs was extracted through manual pestle tissue disruption in TRIzol, followed by DnaseI treatment and purification with the Qiagen RNAeasy Mini purification Kit. Pax7-nGFP+ cells were isolated by FACS directly into cell lysis buffer (RLT) of the Qiagen RNAeasy Plus Micro purification Kit and total RNA extracted according to the kit instructions. cDNA was prepared from 0,4 µg up to 5 µg of total RNA by random-primed reverse transcription (SuperScript III, ThermoFisher 18010093) and real-time PCR was done using SYBR Green Universal Mix (Roche, 13608700) and StepOne-Plus Real Time PCR System (Applied Biosystems). TBP transcript levels were used for normalizations of each target (2ΔCT). At least three biological replicates and technical duplicates were used for each condition method (*Schmittgen et al., 2008*). For SYBR-Green, custom primers were designed using the Primer3Plus online software. Serial dilutions of total cDNA were used ro to calculate the amplification efficiency of each primer set according to the equation: E = 10–1/slope. Primer sequences used are detailed in the Key Resource Table.

## Single-cell qPCR analysis

Gene expression in single cells was analyzed using the Fluidigm Gene Expression Assay (BioMark). Briefly, oesophagus was dissected and digested with trypsin/collagenase to obtain a single cell suspension as described above. Single cells and bulk control (20 cells/well) were sorted directly on a FACS Aria III in 9 µl of Specific Target Amplification (STA) reaction mix from the CellsDirect One-Step qRT-PCR kit (Invitrogen) containing 0.2XTaqMan Gene Expression Assay mix. Pre-amplified cDNA (18 cycles) was obtained according to manufacturer's note and was diluted 1:5 in TE buffer for qPCR. Multiplex qPCR was performed using the microfluidics Biomark system on a Biomark HD for 40 cycles. The same TaqMan probes were used for both RT/STA and qPCR. TaqMan assays used in the study are listed in the Key Resource Table.

### Conversion to relative expression

Raw Ct values were converted in relative expression using the following formula: Log2ex = LOD – Ct [Array] (*Livak et al., 2013*). With the LOD standing for the Limit Of Detection. When the Log2ex value obtained was negative (Ct[array]>LOD) the value was replaced by 0. To set up the LOD, we round the mean of the maximum Ct values for all the genes to the upper limit which gives a LOD of 21.

### Normalization

The resulting relative expression values were normalized to the endogenous controls by subtracting, for each cell, the average of its Actb, Rpl13, Rps29, and Hprt expression levels.

An offset corresponding to the mean of all the calculated means was applied to all obtained values to avoid negative values.

### Single cell filtering

From two independent experiments 66 cells were collected from the esophagus. The criteria to keep a cell for further analysis were the following: i) to discard neurogenic progenitors, cells should not express Pax3 and/or Lhx3. ii) at least 4 out of the five positive control genes should be expressed, as well as at least 2 of the genes of interest. Applying this different filters 23 single cells were selected.

### Correlation coefficient determination and p-value calculation

The Spearman's rho correlation coefficient was calculated using the R function `cor()` with the 'use' parameter set at « `pairwise.complete.obs` », and all the null values previously replaced by

NAs. The coefficient correlation p-value was extracted from the `cor.test()` R function, using the same parameters.

## Data visualization

The heatmap (*Figure 5—figure supplement 2B*) was generated using the pheatmap R package (pheatmap_1.0.10) with default parameters, and the correlogram (*Figure 5F*) was generated using the corrplot R package (corrplot_0.84), with the p-values manually added. Violin plots (*Figure 1—figure supplement 1H,J*) were made in R using the ggplot2 package (ggplot2_3.1.0). R session info: R version 3.5.1 (2018-07-02), platform: x86_64-apple-darwin15.6.0 (64-bit), running under: macOS Sierra 10.12.6.

## Static imaging

Images were acquired using the following systems: Zeiss SteREO Discovery V20 microscope for whole embryos, a Zeiss Axioplan equipped with an Apotome and ZEN software (Carl Zeiss) or a Leica TCS-SP8 with Leica Application Suite (LAS) software for tissue sections and a LSM 700 laser-scanning confocal microscope and ZEN software (Carl Zeiss) for tissue sections and whole mount immunostaining of cleared embryos. All images were assembled in Adobe Photoshop and InDesign (Adobe Systems). Volume-3D rendering of the z-stack series was performed in Imaris (version 7.2.1) software (Bitplane).

## Explant culture

Esophagi from E14.5 *Isl1$^{Cre/+}$:R26$^{mTmG/+}$* embryos were micro-dissected leaving the stomach and pharyngeal muscles attached in RT HBSS (Gibco, 14025). The esophagi were immobilized on individual wells of 8 well glass bottom dishes (Ibidi, 80826) at the stomach and pharyngeal ends using 0.3 µl of Vetbond tissue adhesive (3M, 1469 SB). The explants were immediately embedded in a collagen matrix as previously reported (*Placzek and Dale, 1999*) with slight modifications. 700 µl of collagen-I (Corning, 354236), 200 µl of reconstituted 5X DMEM-F12 (SIGMA, D2906) and 100 µl neutralization buffer (50 mM NaOH, 260 mM NaHCO3, 200 mM Hepes) were mixed and kept on ice. 200 µl of collagen matrix was added to each explant and allowed to polymerize for 10 min in a culture incubator at 37°C, 5% CO$_2$. Explant culture medium was composed of Opti-MEM (Gibco, 51985–026) with 1% P/S and 20% FCS. 250 µl of culture medium containing Met inhibitors or the equivalent amount of DMSO (control) was added to each well and allowed to equilibrate for 30 min in a culture incubator. The Met inhibitors used were MGCD-265 (10 µM, Selleck, 50 mM stock in DMSO) and PF-0417903 (10–20 µM, AbMole, 26.8 mM stock in DMSO).

For static cultures, images of individual wells were acquired at 6 to 12 hr intervals on a Zeiss SteREO Discovery V20 macroscope as Z-stacks and processed with the extended depth focus function on the ZEN software (Carl Zeiss).

For time-lapse imaging, the dish was placed in a microscope incubator chamber (37°C, 5% CO$_2$) and the mGFP signal imaged with the 488 laser on an Leica TCS-SP8 inverted microscope, equipped with a HC PL APO CS2 10X/0.40 dry objective and HyD hybrid detector (496–566 nm). Confocal imaging of optical Z-planes (2.41 µm) were acquired every 15 min over 14 hr using LAS X software. Z-stacks were projected as maximum intensity projection images, stitched and registered (linear registration) in Fiji. The migrating cells were tracked individually frame-by-frame using the 'Manual Tracking' plugin in Fiji. The following parameters were quantitated: total distance (µm, the distance covered by the whole track), velocity (µm/min, ratio between the total distance and total time of the track), displacement (µm, the length of the resultant vector between ti and tf of the track), efficiency (ratio between displacement and total distance), net velocity (µm/min, ratio between the displacement and total time of the track).

## Quantitation of muscle area

The muscle area on transverse esophagus cryosections (*Figure 1—figure supplement 1I*) was quantified on Fiji. Channels were split and threshold levels adjusted on the Tnnt3 channel. The freehand selection tool was used to trace the outline of each esophagus cross section (referred as Region of interest, ROI). Threshold levels were kept constant for all samples. The Analyze/Measure tool was set to calculate the area of the ROI limited to the threshold for the Tnnt3 channel.

## EdU Administration In Vivo

For proliferation experiments in vivo, 5-ethyl-20-deoxyuridine (EdU; Invitrogen E10187) was injected intraperitoneally and detected as described in *Comai et al. (2014)*.

## Acknowledgements

We acknowledge funding support from the Institut Pasteur, Association Française contre le Myopathies, Agence Nationale de la Recherche (Laboratoire d'Excellence Revive, Investissement d'Avenir; ANR-10-LABX-73). We acknowledge the service of Pasteur Imaging platform (PBI), Pasteur Mouse Genetic Engineering platform (CIGM) and Pasteur Flow Cytometry Platform (CEITEC). We also thank C Cimper for technical assistance.

## Additional information

### Funding

| Funder | Grant reference number | Author |
| --- | --- | --- |
| NIH Office of the Director | R01 HD087360 | Mirialys Gallardo<br>Gabrielle Kardon |
| March of Dimes Foundation | | Mirialys Gallardo<br>Gabrielle Kardon |
| Agence Nationale de la Recherche | ANR-10-LABX- 73 | Glenda Comai<br>Eglantine Heude<br>Sebastian Mella<br>Sylvain Paisant<br>Francesca Pala<br>Swetha Gopalakrishnan<br>Shahragim Tajbakhsh |
| Institut Pasteur | | Glenda Comai<br>Eglantine Heude<br>Sebastian Mella<br>Sylvain Paisant<br>Francesca Pala<br>Swetha Gopalakrishnan<br>Shahragim Tajbakhsh |
| Centre National pour la Recherche Scientifique et Technique | | Glenda Comai<br>Eglantine Heude<br>Sebastian Mella<br>Sylvain Paisant<br>Francesca Pala<br>Swetha Gopalakrishnan<br>Shahragim Tajbakhsh |

The funders had no role in study design, data collection and interpretation, or the decision to submit the work for publication.

### Author contributions

Glenda Comai, Eglantine Heude, Conceptualization, Formal analysis, Validation, Investigation, Visualization, Methodology, Writing—original draft, Writing—review and editing; Sebastian Mella, Software, Formal analysis, Visualization, Methodology, Writing—review and editing; Sylvain Paisant, Francesca Pala, Mirialys Gallardo, Gabrielle Kardon, Investigation, Writing—review and editing; Francina Langa, Resources, Investigation, Writing—review and editing; Swetha Gopalakrishnan, Conceptualization, Validation, Investigation, Visualization, Methodology, Writing—original draft, Writing—review and editing; Shahragim Tajbakhsh, Conceptualization, Resources, Supervision, Funding acquisition, Investigation, Project administration, Writing—review and editing

### Author ORCIDs

Glenda Comai  https://orcid.org/0000-0003-3244-3378
Eglantine Heude  https://orcid.org/0000-0003-0563-9607

Gabrielle Kardon  http://orcid.org/0000-0003-2144-4463
Shahragim Tajbakhsh  https://orcid.org/0000-0003-1809-7202

## Ethics

Animal experimentation: Animals were handled as per European Community guidelines and the ethics committee of the Institut Pasteur (CETEA) approved protocols (APAFIS#6354-20160809 l2028839).

## Decision letter and Author response

Decision letter https://doi.org/10.7554/eLife.47460.023
Author response https://doi.org/10.7554/eLife.47460.024

## Additional files

### Supplementary files

• Transparent reporting form
DOI: https://doi.org/10.7554/eLife.47460.021

### Data availability

All data generated or analysed during this study are included in the manuscript and supporting files. Source data files have been provided for Figures 4 and 5.

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
