## [Decision Letter]

Thank you for submitting your article "A distinct cardiopharyngeal mesoderm genetic hierarchy establishes antero-posterior patterning of esophagus muscle" for consideration by *eLife*. Your article has been reviewed by three peer reviewers, including Robert Krauss as the Reviewing Editor and Reviewer #1, and the evaluation has been overseen by Didier Stainier as the Senior Editor. The following individual involved in review of your submission has agreed to reveal their identity:; Carmen Birchmeier (Reviewer #2).

The reviewers have discussed the reviews with one another and the Reviewing Editor has drafted this decision to help you prepare a revised submission.

Summary:

Comai et al. report on genetic requirements for development of esophagus striated muscle (ESM). Previous work had demonstrated that ESM derives from an Isl1+ cardiopharyngeal mesoderm (CPM) population. Although *Isl1* is not a newly identified factor in this process, its involvement had only been shown by lineage tracing; genetic/functional studies were missing due to early lethality of *Isl1* mutants. In this paper, the authors used chimeric mouse embryos to show that *Isl1* and *Tbx1* are required cell-autonomously for myogenic specification of ESM progenitors. They also showed that Met/HGF signaling is required for maintenance, and anterior-to-posterior migration, of a non-differentiating population of ESM progenitors (with Met expressed by ESM progenitors and HGF expressed by esophagus smooth muscle cells). Intriguingly, neck muscles adjacent to the esophagus (pharyngeal and laryngeal) are also CPM-derived but develop normally in the absence of Met/HGF. The data are of high quality.

Essential revisions:

1) In both this paper and an earlier paper (Gopalakrishnan, 2015), the authors have shown that ESM progenitors migrate in an anterior-to-posterior manner to populate the esophageal musculature. As Met/HGF signaling is involved in migration of somitic muscle progenitors into the limb bud this makes sense. However, the expression of HGF by the two layers of esophagus smooth muscle cells raises the question of how this promotes directed, anterior-to-posterior migration (rather than concentrically around the radially-oriented smooth muscles or randomly). HGF mRNA expression "retreats" distally in the limb bud, aiding proximal-distal migration of muscle progenitor cells. A comparison of the position of the front of the migratory cells and HGF expression is critical to understanding whether Met/HGF signals direct the cells or just increase their motility. It would be straightforward for the authors to perform a time-course of qRT-PCR on anterior, middle, and posterior esophagus segments (e.g., at E13.5, E15.5, and E17.5) for HGF and Met expression to address this point. An additional or alternative way to address this question would be to use RNAscope (as in Figure 5) for HGF and Met on longitudinal sections of esophagi at these stages. These experiments would take advantage of techniques the authors already have going, would address an obvious question, and would add results to clarify some of their developmental genetic observations.

2) It is stated, "Analysis on sections revealed that Isl1+ progenitors had seeded the anterior esophagus smooth muscle layers in mutant embryos similarly to controls at E13.5." The number of Isl1+ cells located in the anterior smooth muscle layers in *Met^D/D^* mutants and controls should be provided to back up this statement.

3) As stated in the source data, the graphical representation in Figure 5D-E derives from “190 cells were assessed in total in 5 different E14.5 esophagus sections. Representative of 2 experiments.” This information needs to be added to the legend and expanded upon for clarity.

---

## [Author Response]

Essential revisions:1) In both this paper and an earlier paper (Gopalakrishnan, 2015), the authors have shown that ESM progenitors migrate in an anterior-to-posterior manner to populate the esophageal musculature. As Met/HGF signaling is involved in migration of somitic muscle progenitors into the limb bud this makes sense. However, the expression of HGF by the two layers of esophagus smooth muscle cells raises the question of how this promotes directed, anterior-to-posterior migration (rather than concentrically around the radially-oriented smooth muscles or randomly). HGF mRNA expression "retreats" distally in the limb bud, aiding proximal-distal migration of muscle progenitor cells. A comparison of the position of the front of the migratory cells and HGF expression is critical to understanding whether Met/HGF signals direct the cells or just increase their motility. It would be straightforward for the authors to perform a time-course of qRT-PCR on anterior, middle, and posterior esophagus segments (e.g., at E13.5, E15.5, and E17.5) for HGF and Met expression to address this point. An additional or alternative way to address this question would be to use RNAscope (as in Figure 5) for HGF and Met on longitudinal sections of esophagi at these stages. These experiments would take advantage of techniques the authors already have going, would address an obvious question, and would add results to clarify some of their developmental genetic observations.

The reviewers raise an important point on directed A-P migration of esophagus myogenic progenitors. To address this issue, we have performed the following experiments:

i) RT-qPCR for *Met*, *SMA* and *Hgf* in the anterior, middle and posterior segments of the developing esophagi at E13.5, E15.5, and E17.5. The new data are now presented in Figure 2—figure supplement 1 and developed further in the Results and Discussion sections.

ii) In situRNAscope analysis for *Hgf* on whole-mount esophagi at E13.5 and E14.5, and on transverse sections at E13.5, E14.5 and E16.5. For *Met* analysis, this was done on longitudinal sections at E16.5. We have also performed quantifications of *Hgf* transcripts at the stages indicated above. The new data are shown in Figure 5—figure supplement 1 and discussed in the text.

Our results indicate that the dynamic *Hgf* expression pattern that we observe might contribute to the outer to inner layer and anterior to posterior myogenic cell progression. We have expanded the discussion on how *Hgf* levels might be regulated and might influence myogenic cell migration within the esophagus.

2) It is stated, "Analysis on sections revealed that Isl1+ progenitors had seeded the anterior esophagus smooth muscle layers in mutant embryos similarly to controls at E13.5." The number of Isl1+ cells located in the anterior smooth muscle layers in Met^D/D^ mutants and controls should be provided to back up this statement.

We have now included quantifications of the number of Isl1+ progenitors that have seeded the anterior esophagus of E13.5 control and *Met^D/D^*mutant embryos (Figure 3—figure supplement 1A-A’’). To do so, we enumerated the number of Isl1+ cells in the first 10 consecutive sections of 3 mutant and control embryos. While the total number of Isl1+ cells in the anterior esophagus appeared to be slightly reduced in the mutant, this was not statistically significant. The Results section has been updated accordingly.

3) As stated in the source data, the graphical representation in Figure 5D-E derives from “190 cells were assessed in total in 5 different E14.5 esophagus sections. Representative of 2 experiments.” This information needs to be added to the legend and expanded upon for clarity.

We thank the reviewers for pointing this out. We have now completed our RNAscope analysis to be able to perform statistics on a total of 3 experiments (3 independent embryos). On each experiment, 3-5 sections at the anterior most level of a control esophagus were subjected to RNAscope in situhybridization followed by immunofluorescence. A total of 368 GFP+ cells was assessed for the expression of Myod/Myog and RNAscope score. The overall conclusions remain unchanged. The figure legend and Figure 5—source data 1 have been expanded accordingly.